# Association between normal weight obesity and comorbidities and events of cardiovascular diseases among adults in South China

**Miaomiao Ma**[1,2☯], **Deliang Lv**[1☯], **Xiaobing Wu**[1], **Yuqing Chen**[2], **Shimiao Dai**[2], **Yutian Luo**[3], **Hui Yang**[1], **Wei Xie**[1], **Fengzhu Xie**[1], **Qinggang Shang**[1], **Ziyang Zhang**[1], **Zhiguang Zhao**[1]\*, **Ji-Chang Zhou**[2,4,5]\*

1 Shenzhen Center for Chronic Disease Control, Shenzhen, China, 2 School of Public Health (Shenzhen), Shenzhen Campus of Sun Yat-sen University, Shenzhen, China, 3 School of Public Health, Columbia University, New York, NY, United States of America, 4 Guangdong Province Engineering Laboratory for Nutrition Translation, Guangzhou, Guangdong, China, 5 Guangdong Provincial Key Laboratory of Food, Nutrition and Health, Guangzhou, Guangdong, China

☯ These authors contributed equally to this work.
\* zhoujch8@mail.sysu.edu.cn (JCZ); 1498384005@qq.com (ZZ)

## Abstract

### Background

The increased risks for cardiovascular comorbidities and cardiovascular diseases (CVD) in populations with normal weight obesity (NWO) have not been well-identified. We aimed to study their associations in an adult population in South China.

### Methods

Based on the CVD prevalence of 4% in Shenzhen and a calculated sample size of 6,000, a cross-sectional study with a multi-stage stratified cluster sampling method was conducted in Shenzhen City. The cardiovascular comorbidities being studied were abdominal obesity (AO), diabetes, hypertension, dyslipidemia, metabolic syndrome, and chronic kidney disease, while the CVD events were occurrences of myocardial infarction and strokes. Questionnaire surveys, physical examinations, and laboratory tests were performed. NWO was defined as a condition with the highest tertile of body fat percentage (BF%) among the normal body mass index (BMI) range (18.5–23.9 kg/m$^2$). Continuous data were reported as mean [standard deviation (SD)] and categorical data as percentages (%). CVD comorbidities and CVD events and their detection rates in different groups were compared using ANONA analysis and Chi-squared test. Spearman's correlation coefficients between BF% and cardiometabolic abnormalities were calculated by partial correlation analysis. Multivariate logistic regression models were used to estimate the odds ratios (ORs) and 95% confidence intervals (CIs) for BF%, CVD comorbidities, and CVD events, adjusted for multiple confounders.

ethical review, we did not inform the survey participants that the data would be made freely accessible, which would violate our ethical review policy]. Data are available from the Ethics Committee [contact via Ethics Committee of Shenzhen Center for Chronic Disease Control (via mail xnxgfkk@wjw.sz.gov.cn)] for researchers who meet the criteria for access to confidential data.

**Funding:** This study was supported by the Medical Scientific Research Foundation of Guangdong Province (A2024634) awarded to D.L., the Sanming Project of Medicine in Shenzhen (SZSM202311019) and the Shenzhen Medical Key Discipline Construction Fund (SZXK065) awarded to Zh.Z., the Shenzhen Science and Technology Programs awarded to Q.S. (JCYJ20210324121600003) and J.-C.Z. (JCYJ20200109142446804 and GJHZ20240218114600001), and the National Natural Science Foundation of China (81973038) awarded to J.-C.Z. The funders had no role in study design, data collection and analysis, decision to publish, or preparation of the manuscript.

**Competing interests:** NO authors have competing interests.

## Results

Among the total 6,240 subjects who completed the study and had BMI and BF% data available, 3,086 had normal BMI. The prevalence of NWO was 16.36%, with 13.15% for men and 19.54% for women. With confounders adjusted, the risks of AO (OR = 6.05, 95%CI = 3.40–10.75), essential hypertension (OR = 1.56, 95%CI = 1.09–2.22), dyslipidemia (OR = 1.85, 95%CI = 1.49–2.29), and metabolic syndrome (OR = 4.61, 95%CI = 2.32–9.18) were significantly increased in the populations with NWO compared with the population without NWO ($P$ < 0.05). BF% was not significantly associated with the risk of CVD events in the total (OR = 1.56, 95%CI = 0.83–2.93), male (OR = 1.00, 95%CI = 0.44–2.30), and female populations (OR = 2.53, 95%CI = 0.91–7.06).

## Conclusion

NWO was found to be positively associated with CVD comorbidities but not with CVD events. The current study provides a ground to conduct further studies on whether body fat affects the risk of occurrence of CVD events and the underlying mechanisms in the future.

## 1. Introduction

Obesity is most commonly identified and classified by body mass index (BMI) [1] and has become a major public health issue in the world [2]. Based on normal BMI criteria (18.5 to 23.9 kg/m$^2$) for Chinese populations [3], it was reported that more than half of Chinese adults were either overweight or obese in a most recent national survey [4]. Obesity is closely associated with several comorbidities such as type 2 diabetes, hypertension, coronary artery disease, metabolic syndrome, and many types of cancer [5]. Obese subjects have a 3.5-fold increased likelihood of having hypertension and 60% of hypertension is attributable to an increase in adipose stores [6]. Obesity has also been identified as an independent risk factor for many cancers. Some studies reported that nearly 40% of all cancers can be attributed to overweight and obesity [7]. In particular, endometrial, postmenopausal breast, and colorectal cancers account for over 60% of cancers attributed to obesity [8, 9]. In a pooled meta-analysis of 97 observational studies, Flegal et al. [10] found summary relative risks of all-cause mortality for obesity of 1.18 (95% CI, 1.12–1.25), and obesity is considered to make a great contribution to all-cause mortality. However, because BMI does not distinguish body fat from lean mass and does not reflect true body composition, it has limited accuracy for diagnosing obesity with excess body fat while BMI is within the normal range [11]. A cross-sectional study by Zeng et al. [12] demonstrated that a high body fat percentage (BF%) is a more precise predictor of cardiometabolic risk factors compared to BMI alone. Furthermore, individuals with high BF% despite maintaining a normal BMI may still face an elevated risk of developing cardiovascular diseases (CVDs) [11, 13, 14].

De Lorenzo et al. [15] introduced the term normal weight obesity (NWO) to define individuals who have normal BMI but high BF%. Individuals with NWO often ignore the risk of high BF% since it has no obvious impact on body shape aesthetically. In a meta-analysis including 25 observational studies with 177,792 participants, Mohammadian et al. [2] revealed a correlation between NWO with cardiometabolic risk factors. He found that NWO is correlated with increased odds of dyslipidemia and metabolic syndrome [2]. Furthermore, Gebremedhin et al. [16] conducted a cross-sectional study among 600 adult

participants in Addis Ababa to evaluate the association between NWO with hypertension, elevated blood sugar, and dyslipidemia. He found diastolic blood pressure and odds of hypertension was significantly raised in the NWO individuals compared with normal weight lean [16]. In addition, adults with NWO had elevated blood glucose and increased odds of high blood sugar levels, highlighting the association between NWO and cardiometabolic derangements [16]. People with NWO are likely to develop dyslipidemia [5], insulin resistance [16, 17], changes in blood pressure (BP) [16, 18], and a pro-oxidative status [19]. Thus, to identify whether the condition of NWO is linked to a cardiometabolic risk profile in adults is especially relevant. However, due to their normal BMI and no obvious changes in body shape, they will remain undiagnosed, and no proper preventive measures will be taken until it is too late [2]. It is useful to conduct more studies so that normal weight obesity gets recognition in the public.

There were few epidemiological studies conducted on the prevalence of NWO among Chinese adults and its association with CVD comorbidities and CVD events. In 2018, a national study about NWO and cardiometabolic risks among Chinese [20] showed the prevalence of NWO ranged from 4.52% to 9.68%, and revealed NWO has definite correlations with the risks of hypertension, metabolic syndrome, and cardiovascular diseases. However, its data were collected in 2007 and were not representative of the status quo. There is a need to update the data on the relationship between BF% and CVD comorbidities and CVD events in the BMI-defined NWO population in China. Furthermore, some studies have shown that many types of CVD may have a better prognosis in the overweight or obese population compared to their leaner counterparts, and this phenomenon is referred to as the "obesity paradox" [21, 22]. However, researchers argue that NWO may be associated with a higher risk of CVD events [23]. Since there have been no aggregated meta-analyses on this subject, the relationships between CVD events and NWO remain unclear.

Therefore, in the present study, we aimed to investigate the associations of NWO with comorbidities and events of CVD in a population aged over 18 years in Shenzhen, China.

## 2. Method

### 2.1. Study population

The current investigation was nested in a cross-sectional survey, conducted between April 2021 and January 2022, in Shenzhen, a city with more than 15 million residents in South China. Based on the CVD prevalence of 4% in Shenzhen, the calculated sample size of the simple random sampling was 6,000 in the survey. By using a multi-stage stratified cluster sampling method, a total of 54 communities in 10 districts of Shenzhen were randomly selected as investigation points, and residents who were aged 18 years or older and living in Shenzhen for more than 6 months were randomly selected according to gender and age. The study was approved by the Ethics Committee of Shenzhen Center for Chronic Disease Control (approval no.: SZCCC-2021-007-01-PJ), and all the surveyed subjects signed the written informed consent and were informed about the aims and objectives of the study.

Inclusion criteria for our study were as follows: 1) subjects with a normal BMI of 18.5–23.9 kg/m$^2$ according to "Expert Consensus on Obesity Prevention and Treatment in China" [3]); and 2) provided written and informed consent. Exclusion criteria were as: 1) subjects who came to Shenzhen for sightseeing, visiting relatives, business trips, or short-term training; 2) unable to complete the investigation due to physical illness or refusal to participate in the investigation; and 3) BMI and BF% data were not available.

## 2.2. Measurements

Data were collected with a questionnaire survey, physical examination, and laboratory testing. The questionnaire survey was conducted by uniformly trained investigators, covering information about gender, age, education level, marriage information, ethnic group, smoking, alcohol consumption, physical activity level, history of chronic diseases, etc. In addition, we also collected medication information for antidiabetic, antihypertensive, and/or lipid-regulating purposes from the surveyed subjects. Body height, weight, BF%, waist circumference (WC), and BP were measured by using a standard method. Height was measured using a height meter with an accuracy of 0.1 cm. Weight and BF% were measured using a body composition analyzer (InBody770; Biospace Co., Ltd., Korea) with an accuracy of 0.1 kg. BMI was calculated by dividing weight (kg) by height squared ($m^2$) (keep two decimals). WC was measured using the same brand of measuring tape (length of 1.5 m and accuracy of 0.1 cm) around the abdomen at the midpoint of the line between the anterior superior iliac spine and the lower edge of the twelfth rib at the horizontal level. After participants sat still for 5 minutes, BP was measured twice in the upper right arm using a digital automatic BP monitor (Omron HEM-907, Tokyo, Japan). In addition, according to the international standard, echocardiography and electrocardiogram (ECG) examinations were performed by skilled doctors to evaluate the anatomy and function of the heart and large blood vessels. The physical examination was carried out by uniformly trained investigators. All measuring instruments meet the national metrology certification requirements and were tested and calibrated before using according to the relevant requirements for accuracy and precision.

Venous blood samples were collected in the morning after a 12-h fasting period, and then sent to Shenzhen Chronic Disease Prevention and Control Center for testing fasting blood glucose (FBG), glycated hemoglobin A1c (HbA1c), homocysteine (Hcy), total cholesterol (TC), low-density lipoprotein cholesterol (LDL-C), high-density lipoprotein cholesterol (HDL-C), triacylglycerol (TG), serum creatinine (SCR), and uric acid (UC). The estimated glomerular filtration rate (eGFR) was calculated according to the formula [24]: eGFR = $175 \times$ SCR (mg/dl) $^{-1.234} \times$ age (years) $^{-0.179} \times 0.79$ (if female).

## 2.3. Definitions of NWO, CVD comorbidities, CVD events, and behavioral risks

We used the cut-off value for the highest tertile of BF% in the population with normal BMI to define NWO, since it was reported to be more valid than an arbitrary cut-off not previously validated [25] and there were no established cut-off values for BF% among Asians.

All subjects had a BMI of 18.5 to 24.0 ($\geq 18.5$ to $< 24.0$) kg/$m^2$ and were stratified according to the BF% tertile. For males, the tertile boundaries of BF% were $< 20.5\%$ as low, $20.5\% \leq$ BF% $< 24.6\%$ as medium, and $\geq 24.6\%$ as high; while for females, those of BF% were $< 29.9\%$ as low, $29.9\% \leq$ BF% $< 33.6\%$ as medium, and $\geq 33.6\%$ as high.

The definitions of the studied CVD comorbidities are described below:

1. Abdominal obesity (AO) was defined according to "Expert Consensus on Obesity Prevention and Treatment in China" [3], WC $\geq$ 90 cm for males and WC $\geq$ 85 cm for females.

2. Diabetes, according to "Guideline for the prevention and treatment of type 2 diabetes mellitus in China (2020 edition)" [26], was defined as FBG $\geq$ 7.0 mmol/L and/or blood glucose 2 h after a 75-g oral glucose tolerance test (OGTT) $\geq$ 11.1 mmol/L and/or HbA1c $\geq$ 6.5% in an undiagnosed person or who had a diagnosed history of diabetes.

3. Hypertension [essential hypertension (E-HTN)] was defined as an average systolic blood pressure (SBP) $\geq$ 140 mmHg and/or diastolic blood pressure (DBP) $\geq$ 90 mmHg in an undiagnosed person or who had a diagnosed history of hypertension [27].

4. H-type hypertension (H-HTN) was defined as having a diagnosis of E-HTN with homocysteine (Hcy) $\geq$ 10 μmol/L [28].

5. Dyslipidemia, according to "Chinese guidelines for lipid management (2023)" [29], was defined as TC $\geq$ 6.22 mmol/L and/or LDL-C $\geq$ 4.14 mmol/L and/or HDL-C $\leq$ 1.04 mmol/L and/or TG $\geq$ 2.26 mmol/L in an undiagnosed person or who had a diagnosed history of dyslipidemia.

6. Metabolic syndrome, according to Chinese Diabetes Society criteria (2020) [30], was defined with three or more of the following risk factors: (1) FBG $\geq$ 6.1 mmol/L and/or blood glucose 2 h after an OGTT $\geq$ 7.8 mmol/L and/or diagnosed with diabetes; (2) SBP $\geq$ 130 mmHg and/or DBP $\geq$ 85 mmHg and/or diagnosed with hypertension; (3) plasma TG $\geq$ 1.70 mmol/L or treatment for hypertriglyceridemia; (4) HDL-C $\leq$ 1.04 mmol/L; and (5) AO with WC > 90 cm for males and > 85 cm for females.

7. Chronic kidney disease (CKD) was defined as eGFR < 60 mL/min/1.73 m$^2$ [31].

CVD events include myocardial infarction and stroke in our study.

1. Myocardial infarction was defined as characteristic changes in troponin T and creatine kinase isoenzyme MB (CK-MB) isoform levels, symptoms of myocardial ischemia, changes in electrocardiogram results, or a combination of them [32].

2. Stroke was defined as an acute focal neurological deficit diagnosed by a physician and thought to be of vascular origin (without other cause such as brain tumor) with signs and symptoms lasting $\geq$ 24 hours [33].

CVD events were diagnosed by two trained clinicians based on past medical history, clinical symptoms, ECG, and echocardiography.

Definitions of behavioral/lifestyle factors included alcohol history (drinking over 30 g of alcohol per month for more than 1 year), smoking history (smoking every day or almost every day, with at least 7 cigarettes per week for at least 6 months [32]), and adequate physical activity (participating in moderate intensity of physical activity for at least 150 minutes/week [34]).

## 2.4. Statistical analysis

Continuous data for normally distributed were reported as mean ± standard deviation (SD). ANONA analysis and Dunnett's test were used to compare mean differences between groups (i.e. low vs medium, low vs high, medium vs high). Those continuous data for non-normally distributed were reported as median with interquartile ranges (IQR) (25%, 75%), and nonparametric tests (Kruskal-Wallis test and Bonferroni test) were used to compare mean differences between groups. Categorical data were reported as percentages (%). Chi-squared test and Bonferroni correction were performed to analyze statistical independence between NWO diagnostics results versus cardiovascular comorbidities and CVD outcomes. In addition, Spearman's correlation coefficients between BF% and cardiometabolic abnormalities were calculated by partial correlation analysis adjusted for age, gender, educational level, ethnic group, and statuses of marriage, smoking, and drinking, as well as medication for antidiabetic, antihypertensive, and lipid-regulating purpose on ranks for the specific CVD comorbidity.

Multivariate logistic regression models were used to estimate the odds ratios (ORs) and 95% confidence intervals (CIs) for BF% and CVD comorbidities and CVD events. Two models

were proposed: Model 1 was adjusted for age and gender, while Model 2 was adjusted for potential confounders identified through Spearman correlation analysis.

The restricted cubic spline (RCS) was used to explore the dose-response relationships between BF% levels and studied CVD comorbidities and CVD events [35].

All statistical analyses were performed using SPSS 26.0 and R 4.2.1 software. A two-sided $P$-values $< 0.05$ were considered statistically significant.

# 3. Results

## 3.1. General characteristics

Of the 6,307 subjects who completed the survey, 63 without BMI and BF% data, 4 with a BF% below 5% due to the possible measurement error [20], 2,796 with a BMI $\geq 24$ kg/m$^2$, and 358 with a BMI $< 18.5$ kg/m$^2$ were all excluded. Finally, a total of 3,086 subjects with normal BMI and were included for analysis (S1 Fig in S1 File), and their general characteristics according to BF% tertiles were shown in Table 1.

Among the 3,086 subjects with normal BMI, 39.47% were male. The NWO population consisted of individuals from either gender who were included in the high tertile of BF%. Prevalence of NWO was 16.36% among the total 6,240 subjects with BMI and BF% data available, with 13.15% (408/3,103) and 19.54% (613/3,137) in the males and females respectively. The prevalence of NWO was higher in the females than in the males ($\chi^2 = 46.58$, $P < 0.001$), which suggested that the diagnosis of NWO is dependent on gender.

From the demographic aspect, only age, educational level, and marriage status were associated with BF% in both genders. In the high BF% tertile, the proportions of age above 65 years old, elementary school or below, and divorced and widowed people were all higher than those in the low BF% tertile in both genders. The percentage of adequate physical activity was significantly higher in the medium and high BF% tertiles compared with the low BF% tertile only in the males. Compared with the low BF% tertile, anthropometric parameters (weight, WC, and BMI), SBP, DBP, FBG, blood lipids (TC, LDL-C, and TG), and UA significantly increased in the medium and high BF% tertiles of both genders ($P < 0.05$), while eGFR had no significant correlation with BF% in both genders. In addition, HbA1c and Hcy increased with BF% tertile only in males ($P < 0.05$).

## 3.2. Correlation between cardiometabolic risk factors and BF%

We also evaluated the association between BF% as a continuous variable and markers of metabolic diseases (Table 2). After adjusting for age, gender, educational level, ethnic group, status of marriage, smoking and drinking, and medication for antidiabetic, antihypertensive, and/or lipid-regulating purposes. We found that BF% was positively correlated with WC, SBP, DBP, Hcy, TC, LDL-C, TG, UA, and eGFR levels, and negatively correlated with HDL-C and SCR levels. There were no correlations between BF% and FBG and HbA1c levels.

## 3.3. Prevalences of comorbidities and events of CVD among different BF% tertiles

We evaluated the prevalences of CVD comorbidities and CVD events according to gender-specific BF% tertiles (Fig 1 and S1 Table in S1 File). The estimated prevalences of CVD comorbidities (AO, diabetes, E-HTN, H-HTN, dyslipidemia, metabolic syndrome, and CKD) and CVD events in the total populations were all higher in the high BF% tertile (NWO) than the low and medium BF% tertiles ($P < 0.05$), and gradually increased from the low, medium, to high BF% tertile.

**Table 1. General characteristics of study subjects grouped by body fat percentage (BF%) tertile.**

| | Male | | | P | Female | | | P |
|---|---|---|---|---|---|---|---|---|
| | Low | Medium | High | | Low | Medium | High | |
| N (%) | 403 (33.09) | 407 (33.42) | 408 (33.50) | | 621 (33.33) | 629 (33.76) | 613 (32.90) | |
| Age, y | 33.0 [25.0, 41.0] | 34.0 [29.0, 47.0] [a] | 38.0 [31.0, 54.0] [a, b] | < 0.001 | 34.0 [28.0, 42.0] | 35.0 [28.0, 45.0] [a] | 36.0 [29.0, 50.0] [a, b] | < 0.001 |
| Age group, n (%) | | | | | | | | |
| 18–24 y | 89 (22.08) | 48 (11.79) | 35 (8.58) | < 0.001 | 83 (13.30) | 94 (14.90) | 77 (12.56) | < 0.001 |
| 25–34 y | 152 (37.72) | 159 (39.07) | 134 (32.84) | | 253 (40.54) | 214 (33.91) | 199 (32.46) | |
| 35–44 y | 80 (19.85) | 84 (20.64) | 88 (21.57) | | 156 (25.00) | 155 (24.56) | 129 (21.04) | |
| 45–54 y | 46 (11.41) | 54 (13.27) | 55 (13.48) | | 73 (11.70) | 91 (14.42) | 87 (14.19) | |
| 55–64 y | 24 (5.96) | 38 (9.34) | 36 (8.82) | | 35 (5.61) | 42 (6.66) | 53 (8.65) | |
| ≥ 65 y | 12 (2.98) | 24 (5.90) | 60 (14.71) | | 24 (3.85) | 35 (5.55) | 68 (11.09) | |
| Educational level, n (%) | | | | | | | | |
| elementary school or below | 20 (4.96) | 18 (4.42) | 33 (8.09) | 0.041 | 40 (6.41) | 45 (7.13) | 65 (10.60) | 0.033 |
| middle school | 84 (20.84) | 65 (15.97) | 84 (20.59) | | 118 (18.91) | 133 (21.08) | 140 (22.84) | |
| high school | 101 (25.06) | 109 (26.78) | 114 (27.94) | | 114 (18.27) | 114 (18.07) | 110 (17.94) | |
| college or higher | 198 (49.13) | 215 (52.83) | 177 (43.38) | | 352 (56.41) | 339 (53.72) | 298 (48.61) | |
| Ethnic, Han, n (%) | 374 (92.80) | 391 (96.07) | 401 (98.28) | 0.001 | 601 (96.31) | 602 (95.40) | 574 (93.64) | 0.085 |
| Marriage, n (%) | | | | | | | | |
| unmarried | 151 (37.47) | 102 (25.06) | 87 (21.32) | < 0.001 | 149 (23.88) | 160 (25.36) | 125 (20.39) | 0.047 |
| married or remarried | 243 (60.30) | 288 (70.76) | 311 (76.23) | | 448 (71.79) | 453 (71.79) | 453 (73.90) | |
| divorced or widowed | 9 (2.23) | 17 (4.18) | 10 (2.45) | | 27 (4.33) | 18 (2.85) | 35 (5.71) | |
| Smoking, n (%) | 150 (37.22) | 162 (39.80) | 139 (34.07) | 0.237 | 10 (1.60) | 6 (0.95) | 8 (1.31) | 0.591 |
| Drinking, n (%) | 223 (55.33) | 218 (53.56) | 216 (52.94) | 0.778 | 131 (20.99) | 133 (21.08) | 102 (16.64) | 0.080 |
| Adequate PA, n (%) | 158 (40.62) | 119 (30.75) | 121 (30.95) | 0.004 | 240 (41.03) | 240 (41.03) | 221 (39.05) | 0.733 |
| Antidiabetic drug, n (%) | 9 (2.24) | 18 (4.43) | 19 (4.66) | 0.138 | 11 (1.77) | 20 (3.18) | 12 (1.96) | 0.197 |
| Antihypertensive drug, n (%) | 14 (3.47) | 26 (6.39) | 46 (11.27) [a, b] | < 0.001 | 26 (4.17) | 40 (6.34) | 36 (5.87) | 0.205 |
| Lipid drug, n (%) | 4 (0.99) | 3 (0.74) | 9 (2.21) | 0.144 | 8 (1.28) | 14 (2.22) | 19 (3.10) | 0.093 |
| Height, cm | 169.97 ± 6.50 | 169.21 ± 6.19 | 166.71 ± 6.73 [a, b] | < 0.001 | 158.03 ± 5.90 | 157.57 ± 5.92 | 155.58 ± 5.84 [a, b] | < 0.001 |
| Weight, kg | 60.42 ± 6.35 | 63.04 ± 6.10 [a] | 63.10 ± 5.90 [a] | < 0.001 | 51.10 ± 5.01 | 53.40 ± 5.11 [a] | 54.13 ± 4.91 [a, b] | < 0.001 |
| WC, cm | 76.65 ± 5.43 | 81.22 ± 5.03 [a] | 83.40 ± 5.47 [a, b] | < 0.001 | 71.51 ± 5.65 | 74.02 ± 7.38 [a] | 76.47 ± 6.89 [a, b] | < 0.001 |
| BMI, kg/m$^2$ | 20.88 ± 1.46 | 21.98 ± 1.30 [a] | 22.67 ± 1.05 [a, b] | < 0.001 | 20.43 ± 1.28 | 21.48 ± 1.36 [a] | 22.33 ± 1.20 [a, b] | < 0.001 |
| BF, % | 17.12 ± 2.67 | 22.60 ± 1.18 [a] | 27.79 ± 2.96 [a, b] | < 0.001 | 26.62 ± 2.75 | 31.92 ± 1.04 [a] | 36.19 ± 2.06 [a, b] | < 0.001 |
| SBP, mmHg | 120.71 ± 12.76 | 123.98 ± 13.40 [a] | 126.30 ± 14.51 [a, b] | < 0.001 | 112.22 ± 12.78 | 113.36 ± 14.78 | 116.87 ± 16.17 [a, b] | < 0.001 |
| DBP, mmHg | 75.89 ± 9.10 | 78.92 ± 9.09 [a] | 80.84 ± 9.59 [a, b] | < 0.001 | 71.82 ± 8.66 | 72.42 ± 8.16 | 74.14 ± 8.59 [a, b] | < 0.001 |
| FBG, mmol/L | 5.00 [4.70, 5.30] | 5.10 [4.80, 5.50] [a] | 5.20 [4.80, 5.50] [a, b] | < 0.001 | 4.90 [4.70, 5.30] | 5.00 [4.70, 5.30] [a] | 5.00 [4.80, 5.40] [a] | < 0.001 |
| HbA1c, % | 5.34 ± 0.60 | 5.42 ± 0.64 | 5.49 ± 0.85 | 0.007 | 5.30 ± 0.50 | 5.30 ± 0.53 | 5.35 ± 0.57 | 0.221 |
| Hcy, μmol/L | 13.10 [11.40, 15.75] | 13.70 [11.80, 16.40] | 13.80 [11.80, 13.38] [a] | 0.013 | 9.70 [8.40, 11.60] | 9.80 [8.40, 11.60] | 10.00 [8.50, 12.10] | 0.159 |
| TC, mmol/L | 4.86 ± 0.99 | 5.10 ± 0.98 [a] | 5.15 ± 0.98 [a] | < 0.001 | 4.93 ± 1.06 | 4.95 ± 1.09 | 5.14 ± 1.13 [a, b] | 0.001 |
| LDL-C, mmol/L | 3.15 ± 0.74 | 3.36 ± 0.73 [a] | 3.43 ± 0.74 [a] | < 0.001 | 3.09 ± 0.76 | 3.16 ± 0.76 | 3.34 ± 0.84 [a, b] | < 0.001 |
| HDL-C, mmol/L | 1.40 ± 0.30 | 1.32 ± 0.27 [a] | 1.26 ± 0.25 [a, b] | < 0.001 | 1.57 ± 0.32 | 1.48 ± 0.31 [a] | 1.48 ± 0.29 [a] | < 0.001 |
| TG, mmol/L | 1.03 [0.78, 1.39] | 1.22 [0.88, 1.76] [a] | 1.38 [1.04, 2.03] [a, b] | < 0.001 | 0.86 [0.68, 1.16] | 0.95 [0.74, 1.31] [a] | 1.10 [0.81, 1.46] [a, b] | < 0.001 |
| SCR, μmol/L | 83.54 ± 12.21 | 82.25 ± 12.52 | 81.00 ± 14.58 [a] | 0.023 | 60.49 ± 8.68 | 59.45 ± 9.91 | 58.99 ± 12.65 [a] | 0.037 |
| UA, μmol/L | 390.43 ± 77.88 | 409.31 ± 82.01 [a] | 415.29 ± 88.48 [a] | < 0.001 | 293.78 ± 65.07 | 298.86 ± 70.11 | 311.32 ± 73.10 [a, b] | < 0.001 |

(*Continued*)

**Table 1.** (Continued)

| | Male | | | | Female | | | |
|---|---|---|---|---|---|---|---|---|
| | Low | Medium | High | *P* | Low | Medium | High | *P* |
| eGFR, ml/min | 105.64 [91.84, 114.97] | 103.45 [92.79, 114.34] | 104.08 [90.78, 113.59] | 0.335 | 112.44 [100.83, 119.95] | 113.58 [101.47, 121.17] | 112.92 [98.60, 121.82] | 0.488 |

BMI, body mass index; DBP, diastolic blood pressure; eGFR, estimated glomerular filtration rate; FBG, fasting blood glucose; HbA1c, glycated hemoglobin A1c; Hcy, homocysteine; HDL-C, high-density lipoprotein; LDL-C, low-density lipoprotein; PA, physical activity; SBP, systolic blood pressure; SCR, serum, creatinine; TC, total cholesterol; TG, triacylglycerol; UA, uric acid; WC, waist circumference.

Categorical data are expressed as *n* (%). Continuous data are expressed as mean ± SD for normally distributed data or median [IQR] for non-normally distributed data.

All subjects represented had a BMI of 18.5 to 24.0 kg/m$^2$ and were stratified according to tertile [male: tertile boundaries were as low (BF% < 20.5%), medium (20.5% ≤ BF% < 24.6%), and high (BF% ≥ 24.6%); female: low (BF% < 29.9%), medium (29.9% ≤ BF% < 33.6%), and high (BF% ≥ 33.6%).

[a] *P* < 0.05 vs low

[b] *P* < 0.05 vs medium.

We also analyzed the effect of gender on the prevalences of CVD comorbidities and CVD events. In the males, compared with the low BF% tertile, the prevalences of CVD comorbidities and events were all higher in high BF% tertile (*P* < 0.05), but compared with the medium BF% tertile, the prevalence of diabetes, dyslipidemia, CKD, and CVD events were not higher in the high BF% tertile. In the females, except for diabetes and CKD, the prevalences of AO, E-HTN, H-HTN, dyslipidemia, metabolic syndrome, and CVD events were all higher in high BF% tertile than in the low BF% tertile, but compared with the second BF% tertile, only the prevalence of E-HTN was higher in the high BF% tertile.

**Table 2. Correlations between metabolic parameters and BF% as a continuous variable.**

| Variables | *r* | *P* | *r'* | *P'* |
|---|---|---|---|---|
| WC (cm) | - 0.048 | 0.008 | 0.343 | < 0.001 |
| FBG (mmol/L) | - 0.008 | 0.672 | 0.032 | 0.077 |
| HbA1c (%) | - 0.004 | 0.804 | 0.007 | 0.683 |
| SBP (mmHg) | - 0.105 | < 0.001 | 0.080 | < 0.001 |
| DBP (mmHg) | - 0.097 | < 0.001 | 0.123 | < 0.001 |
| Hcy (μmol/L) | - 0.192 | < 0.001 | 0.051 | 0.005 |
| TC (mmol/L) | 0.079 | < 0.001 | 0.087 | < 0.001 |
| LDL-C (mmol/L) | 0.070 | < 0.001 | 0.132 | < 0.001 |
| HDL-C (mmol/L) | 0.090 | < 0.001 | - 0.169 | < 0.001 |
| TG (mmol/L) | 0.002 | 0.925 | 0.156 | < 0.001 |
| SCR (μmol/L) | - 0.508 | < 0.001 | - 0.0095 | < 0.001 |
| UA (μmol/L) | - 0.307 | < 0.001 | 0.139 | < 0.001 |
| eGFR (ml/min) | 0.108 | < 0.001 | 0.103 | < 0.001 |

BF%, body fat percentage; DBP, diastolic blood pressure; eGFR, estimated glomerular filtration rate; FBG, fasting blood glucose; HbA1c, glycated hemoglobin A1c; Hcy, homocysteine; HDL-C, high-density lipoprotein; LDL-C, low-density lipoprotein; SBP, systolic blood pressure; SCR, serum, creatinine; TC, total cholesterol; TG, triacylglycerol; UA, uric acid; WC, waist circumference.

*r* and *P* were crude correlation indexes. *r'* and *P'* were adjusted for age, gender, educational level, ethnic group, physical activity, and statuses of marriage, smoking, and drinking; in addition, the histories of using antidiabetic, antihypertensive, and lipid-regulating drugs were respectively adjusted for FBG and HbA1c, SBP and DBP, and TC, HDL-C, LDL-C, and TG.

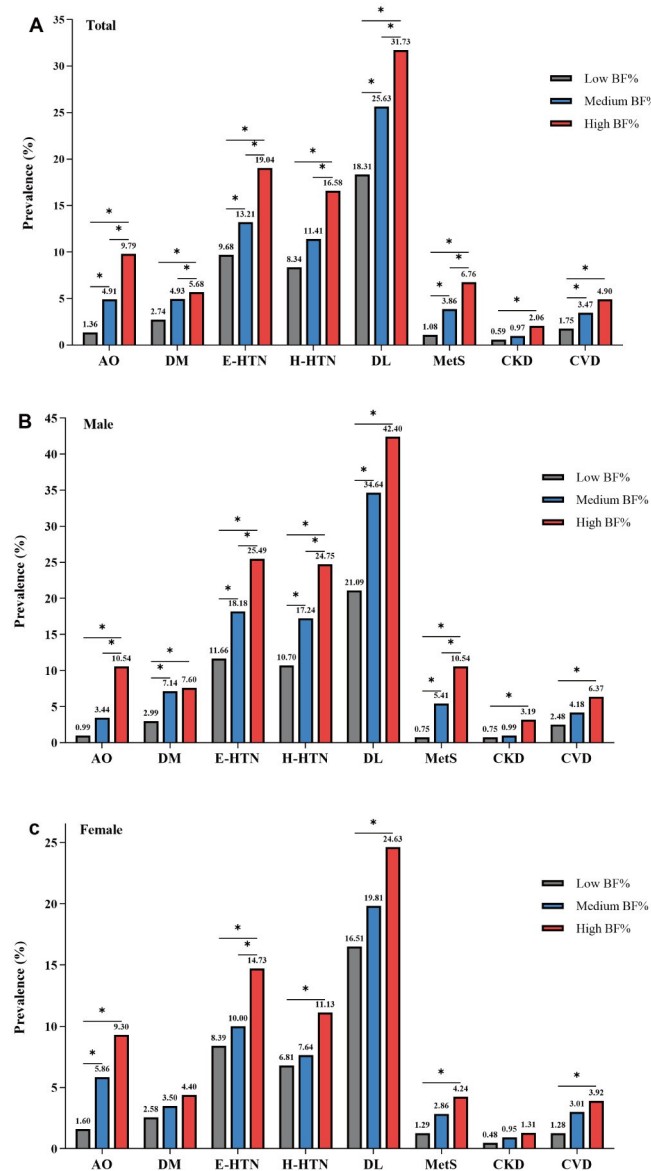

**Fig 1.** Prevalence (%) of comorbidities and events of cardiovascular diseases in the total (A), male (B), and female populations (C) among different tertile BF% groups. AO, abdominal obesity; BF%, body fat percentage; CKD, chronic kidney disease; CVD, cardiovascular disease; DM, diabetic mellitus (diabetes); DL, dyslipidemia; E-HTN, essential hypertension; H-HTN, H-type hypertension; MetS, metabolic syndrome. $*P < 0.05$.

## 3.4. Associations of BF% with comorbidities and events of CVD

Figs 2–4 and S2 and S3 Tables in S1 File show the crude and adjusted ORs of the comorbidities and events of CVD and their associations with NWO

Among total populations (Fig 2 and S2 Table in S1 File), the crude ORs of CVD comorbidities and CVD events in high BF% tertile were significantly higher compared with the low BF% tertile. In both models, the CVD comorbidities of AO (OR$_1$ [for OR in Model 1] = 6.02, 95% CI$_1$ = 3.39–10.69; OR$_2$ [for OR in Model 2] = 6.05, 95%CI$_2$ = 3.40–10.75), E-HTN (OR$_1$ = 1.36, 95%CI$_1$ = 1.01–1.84; OR$_2$ = 1.56, 95%CI$_2$ = 1.09–2.22), dyslipidemia (OR$_1$ = 1.81, 95%CI$_1$ = 1.46–2.25; OR$_2$ = 1.85, 95%CI$_2$ = 1.49–2.29), and metabolic syndrome (OR$_1$ = 4.06, 95%CI$_1$ =

**Total Population**

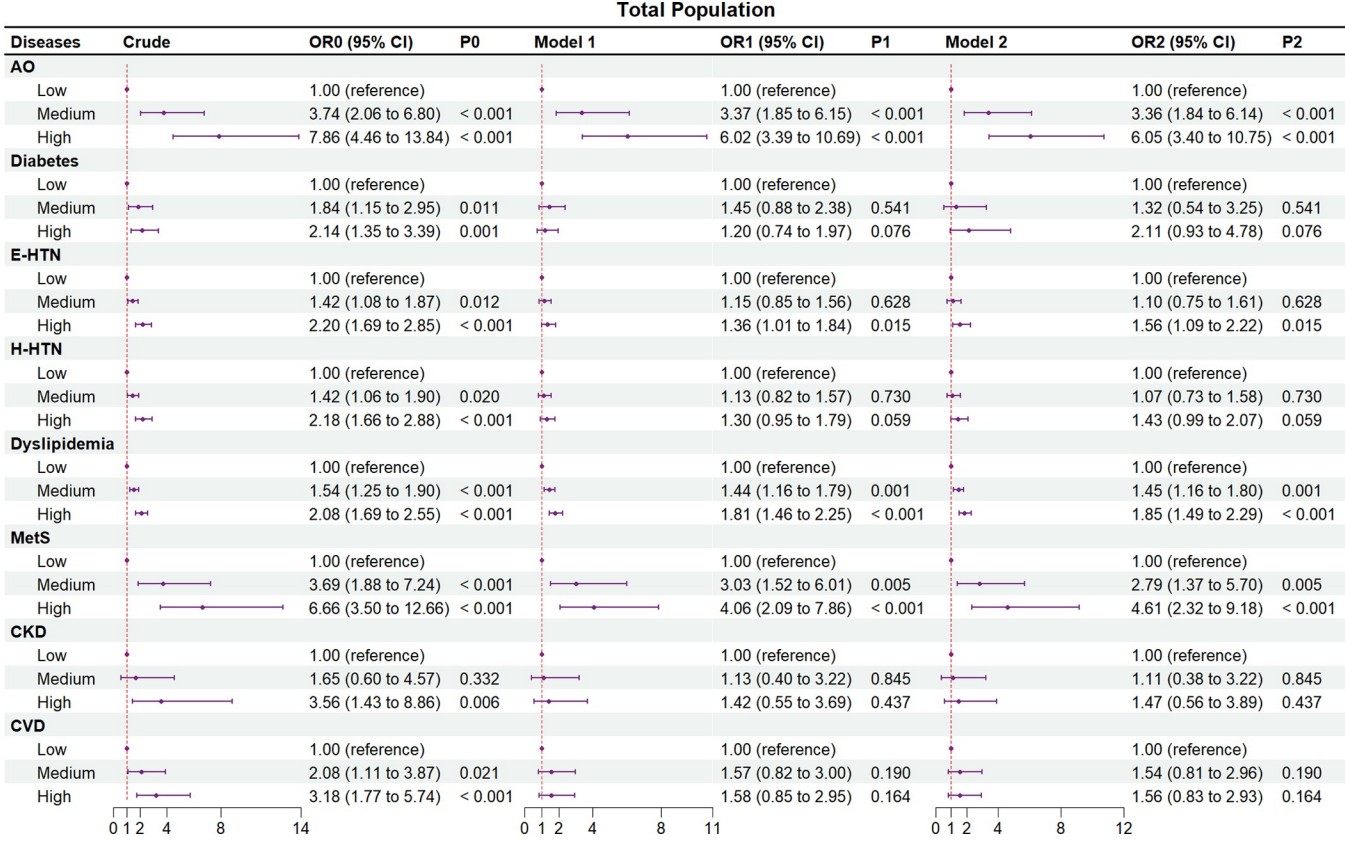

| Diseases | Crude | OR0 (95% CI) | P0 | Model 1 | OR1 (95% CI) | P1 | Model 2 | OR2 (95% CI) | P2 |
|---|---|---|---|---|---|---|---|---|---|
| **AO** | | | | | | | | | |
| Low | | 1.00 (reference) | | | 1.00 (reference) | | | 1.00 (reference) | |
| Medium | | 3.74 (2.06 to 6.80) | < 0.001 | | 3.37 (1.85 to 6.15) | < 0.001 | | 3.36 (1.84 to 6.14) | < 0.001 |
| High | | 7.86 (4.46 to 13.84) | < 0.001 | | 6.02 (3.39 to 10.69) | < 0.001 | | 6.05 (3.40 to 10.75) | < 0.001 |
| **Diabetes** | | | | | | | | | |
| Low | | 1.00 (reference) | | | 1.00 (reference) | | | 1.00 (reference) | |
| Medium | | 1.84 (1.15 to 2.95) | 0.011 | | 1.45 (0.88 to 2.38) | 0.541 | | 1.32 (0.54 to 3.25) | 0.541 |
| High | | 2.14 (1.35 to 3.39) | 0.001 | | 1.20 (0.74 to 1.97) | 0.076 | | 2.11 (0.93 to 4.78) | 0.076 |
| **E-HTN** | | | | | | | | | |
| Low | | 1.00 (reference) | | | 1.00 (reference) | | | 1.00 (reference) | |
| Medium | | 1.42 (1.08 to 1.87) | 0.012 | | 1.15 (0.85 to 1.56) | 0.628 | | 1.10 (0.75 to 1.61) | 0.628 |
| High | | 2.20 (1.69 to 2.85) | < 0.001 | | 1.36 (1.01 to 1.84) | 0.015 | | 1.56 (1.09 to 2.22) | 0.015 |
| **H-HTN** | | | | | | | | | |
| Low | | 1.00 (reference) | | | 1.00 (reference) | | | 1.00 (reference) | |
| Medium | | 1.42 (1.06 to 1.90) | 0.020 | | 1.13 (0.82 to 1.57) | 0.730 | | 1.07 (0.73 to 1.58) | 0.730 |
| High | | 2.18 (1.66 to 2.88) | < 0.001 | | 1.30 (0.95 to 1.79) | 0.059 | | 1.43 (0.99 to 2.07) | 0.059 |
| **Dyslipidemia** | | | | | | | | | |
| Low | | 1.00 (reference) | | | 1.00 (reference) | | | 1.00 (reference) | |
| Medium | | 1.54 (1.25 to 1.90) | < 0.001 | | 1.44 (1.16 to 1.79) | 0.001 | | 1.45 (1.16 to 1.80) | 0.001 |
| High | | 2.08 (1.69 to 2.55) | < 0.001 | | 1.81 (1.46 to 2.25) | < 0.001 | | 1.85 (1.49 to 2.29) | < 0.001 |
| **MetS** | | | | | | | | | |
| Low | | 1.00 (reference) | | | 1.00 (reference) | | | 1.00 (reference) | |
| Medium | | 3.69 (1.88 to 7.24) | < 0.001 | | 3.03 (1.52 to 6.01) | 0.005 | | 2.79 (1.37 to 5.70) | 0.005 |
| High | | 6.66 (3.50 to 12.66) | < 0.001 | | 4.06 (2.09 to 7.86) | < 0.001 | | 4.61 (2.32 to 9.18) | < 0.001 |
| **CKD** | | | | | | | | | |
| Low | | 1.00 (reference) | | | 1.00 (reference) | | | 1.00 (reference) | |
| Medium | | 1.65 (0.60 to 4.57) | 0.332 | | 1.13 (0.40 to 3.22) | 0.845 | | 1.11 (0.38 to 3.22) | 0.845 |
| High | | 3.56 (1.43 to 8.86) | 0.006 | | 1.42 (0.55 to 3.69) | 0.437 | | 1.47 (0.56 to 3.89) | 0.437 |
| **CVD** | | | | | | | | | |
| Low | | 1.00 (reference) | | | 1.00 (reference) | | | 1.00 (reference) | |
| Medium | | 2.08 (1.11 to 3.87) | 0.021 | | 1.57 (0.82 to 3.00) | 0.190 | | 1.54 (0.81 to 2.96) | 0.190 |
| High | | 3.18 (1.77 to 5.74) | < 0.001 | | 1.58 (0.85 to 2.95) | 0.164 | | 1.56 (0.83 to 2.93) | 0.164 |

**Fig 2. Associations of normal weight obesity with comorbidities and events of cardiovascular disease among all subjects.** AO, abdominal obesity; BF%, body fat percentage; CI, confidence interval; CKD, chronic kidney disease; CVD, cardiovascular disease; E-HTN, essential hypertension; H-HTN, H-type hypertension; MetS, metabolic syndrome; OR, odds ratio; Ref, reference. All subjects represented had a body mass index of 18.5 to 24.0 kg/m² and were stratified according to tertile [male: tertile boundaries were as low (BF% < 20.5%), medium (20.5% ≤ BF% < 24.6%) and high (BF% ≥ 24.6%); female: low (BF % < 29.9%), medium (29.9% ≤ BF% < 33.6%) and high (BF% ≥ 33.6%)]. Crude, without adjustment for other risk factors. Model 1, adjusted for age and gender. Model 2, adjusted for age, gender, educational level, ethnic group, and statuses of marriage, smoking, and drinking. In addition, the histories of using antidiabetic, antihypertensive, and lipid-regulating drugs were respectively adjusted for diabetes, E-HTN and H-HTN, and dyslipidemia, and the history of using more than three drugs was adjusted for metabolic syndrome.

2.09–7.86; $OR_2$ = 4.61, 95%$CI_2$ = 2.32–9.18) were all significantly higher in the high BF% tertile compared with the low BF% tertile ($P < 0.05$). In both models, diabetes ($OR_1$ = 1.20, 95%$CI_1$ = 0.74–1.97; $OR_2$ = 2.11, 95%$CI_2$ = 0.93–4.78), H-HTN ($OR_1$ = 1.30, 95%$CI_1$ = 0.95–1.79; $OR_2$ = 1.43, 95%$CI_2$ = 0.99–2.07), and CKD ($OR_1$ = 1.42, 95%$CI_1$ = 0.55–3.69; $OR_2$ = 1.47, 95%$CI_2$ = 0.56–3.89) were suggested to have no significant association with BF% ($P > 0.05$). We also found the ORs of AO, E-HTN, dyslipidemia, and metabolic syndrome increased from the medium to the high BF% tertile. The models showed that BF% was not significantly associated with the risk of CVD events ($OR_1$ = 1.58, 95%$CI_1$ = 0.85–2.95; $OR_2$ = 1.56, 95%$CI_2$ = 0.83–2.93).

We also performed a subgroup analysis based on gender (Figs 3, 4 and S3 Table in S1 File). In the males (Fig 3 and S3 Table in S1 File), except diabetes ($OR_1$ = 1.48, 95%$CI_1$ = 0.72–3.07; $OR_2$ = 2.45, 95%$CI_2$ = 0.63–9.50) and CKD ($OR_1$ = 1.78, 95%$CI_1$ = 0.47–6.69; $OR_2$ = 1.65, 95% $CI_2$ = 0.42–6.47), AO ($OR_1$ = 9.88, 95%$CI_1$ = 3.47–28.11; $OR_2$ = 8.98, 95%$CI_2$ = 3.14–25.70), E-HTN ($OR_1$ = 1.66, 95%$CI_1$ = 1.10–2.52; $OR_2$ = 1.66, 95%$CI_2$ = 1.02–2.70), H-HTN ($OR_1$ = 1.77, 95%$CI_1$ = 1.16–2.71; $OR_2$ = 1.83, 95%$CI_2$ = 1.11–3.01), dyslipidemia ($OR_1$ = 2.56, 95%$CI_1$ = 1.86–3.53; $OR_2$ = 2.73, 95%$CI_2$ = 1.97–3.79), and metabolic syndrome ($OR_1$ = 12.35, 95%$CI_1$

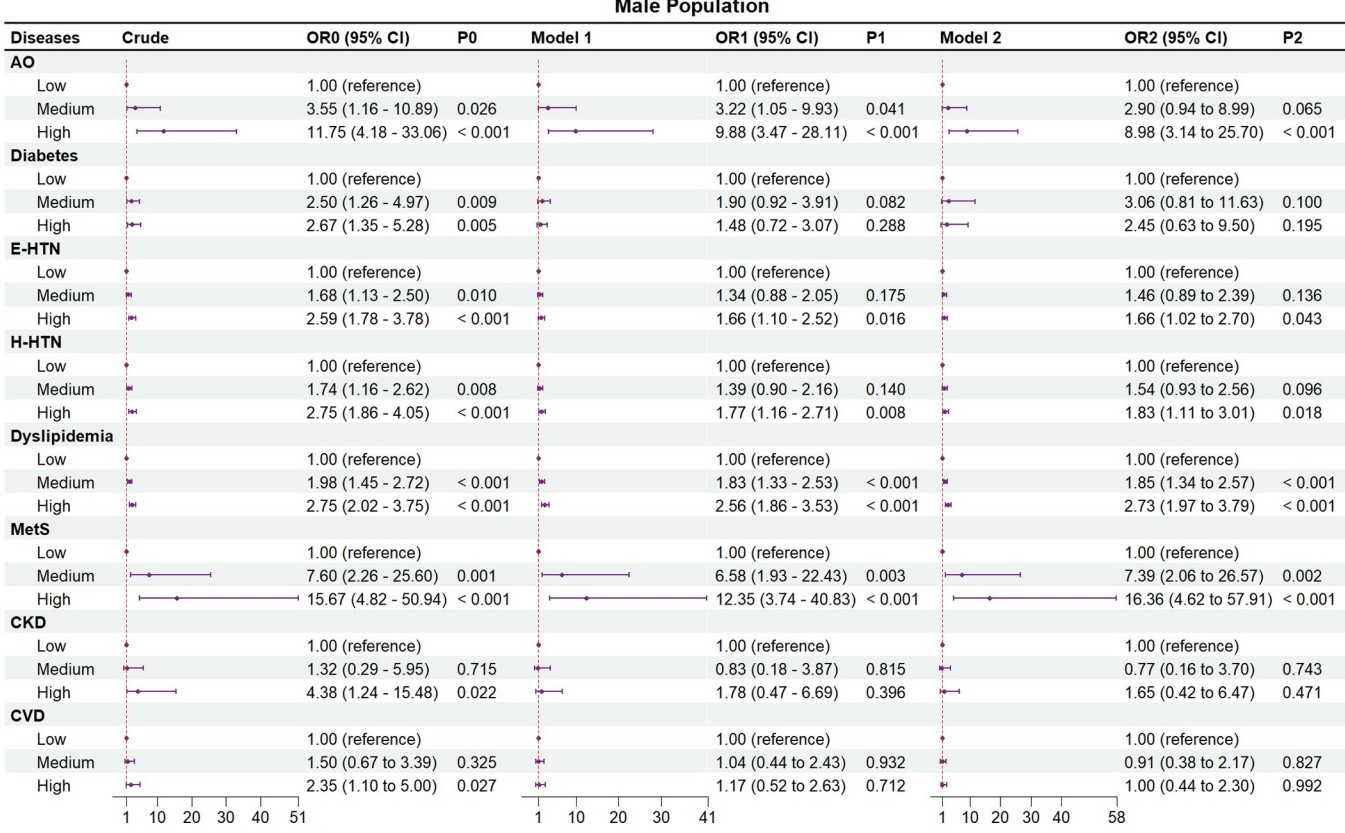

**Fig 3. Associations of normal weight obesity with comorbidities and events of cardiovascular disease among male subjects.** AO, abdominal obesity; BF%, body fat percentage; CI, confidence interval; CKD, chronic kidney disease; CVD, cardiovascular disease; E-HTN, essential hypertension; H-HTN, H-type hypertension; MetS, metabolic syndrome; OR, odds ratio; Ref, reference. Crude, without adjustment for other risk factors. Model 1, adjusted for age and gender. Model 2, adjusted for age, gender, educational level, ethnic group, and statuses of marriage, smoking, and drinking. In addition, the histories of using antidiabetic, antihypertensive, and lipid-regulating drugs were respectively adjusted for diabetes, E-HTN and H-HTN, and dyslipidemia, and the history of using more than three drugs was adjusted for metabolic syndrome.

= 3.74–40.83; $OR_2$ = 16.63, 95%$CI_2$ = 4.62–57.91) were all significantly higher in the high BF% tertile compared with the low BF% tertile ($P < 0.05$) in the two models. In the females (Fig 4 and S3 Table in S1 File), compared with the low BF% tertile, only AO ($OR_1$ = 4.70, 95%$CI_1$ = 2.35–9.43; $OR_2$ = 4.86, 95%$CI_2$ = 2.42–9.76), and dyslipidemia ($OR_1$ = 1.37, 95%$CI_1$ = 1.02–1.84; $OR_2$ = 1.38, 95%$CI_2$ = 1.03–1.85) were found to be significantly higher in high BF% tertile ($P < 0.05$) in both Model 1 and Model 2, while diabetes ($OR_1$ = 1.01, 95%$CI_1$ = 0.52–1.97; $OR_2$ = 2.21, 95%$CI_2$ = 0.77–6.33), E-HTN ($OR_1$ = 1.11, 95%$CI_1$ = 0.72–1.71; $OR_2$ = 1.51, 95%$CI_2$ = 0.88–2.58), H-HTN ($OR_1$ = 0.89, 95%$CI_1$ = 0.55–1.44; $OR_2$ = 1.09, 95%$CI_2$ = 0.62–1.92), metabolic syndrome ($OR_1$ = 1.66, 95%$CI_1$ = 0.71–3.92; $OR_2$ = 2.06, 95%$CI_2$ = 0.83–5.10), and CKD ($OR_1$ = 1.18, 95%$CI_1$ = 0.29–4.71; $OR_2$ = 1.27, 95%$CI_2$ = 0.31–5.24) were not significantly higher ($P > 0.05$) either in Model 1 or Model 2. Furthermore, both the male ($OR_1$ = 1.17, 95%$CI_1$ = 0.52–2.63; $OR_2$ = 1.00, 95%$CI_2$ = 0.44–2.30) and female ($OR_1$ = 2.56, 95%$CI_1$ = 0.93–7.05; $OR_2$ = 2.53, 95%$CI_2$ = 0.91–7.06) subgroups exhibited similar findings to the overall populations, indicating that the risk of CVD events was not significantly higher in the high BF% tertile compared to the lower BF% tertiles.

To further explore the increased CVD comorbidities and CVD events associated with active interventions in the NWO population, we excluded subjects who were self-reported to have hypertension, diabetes, and dyslipidemia for further analysis. After excluding

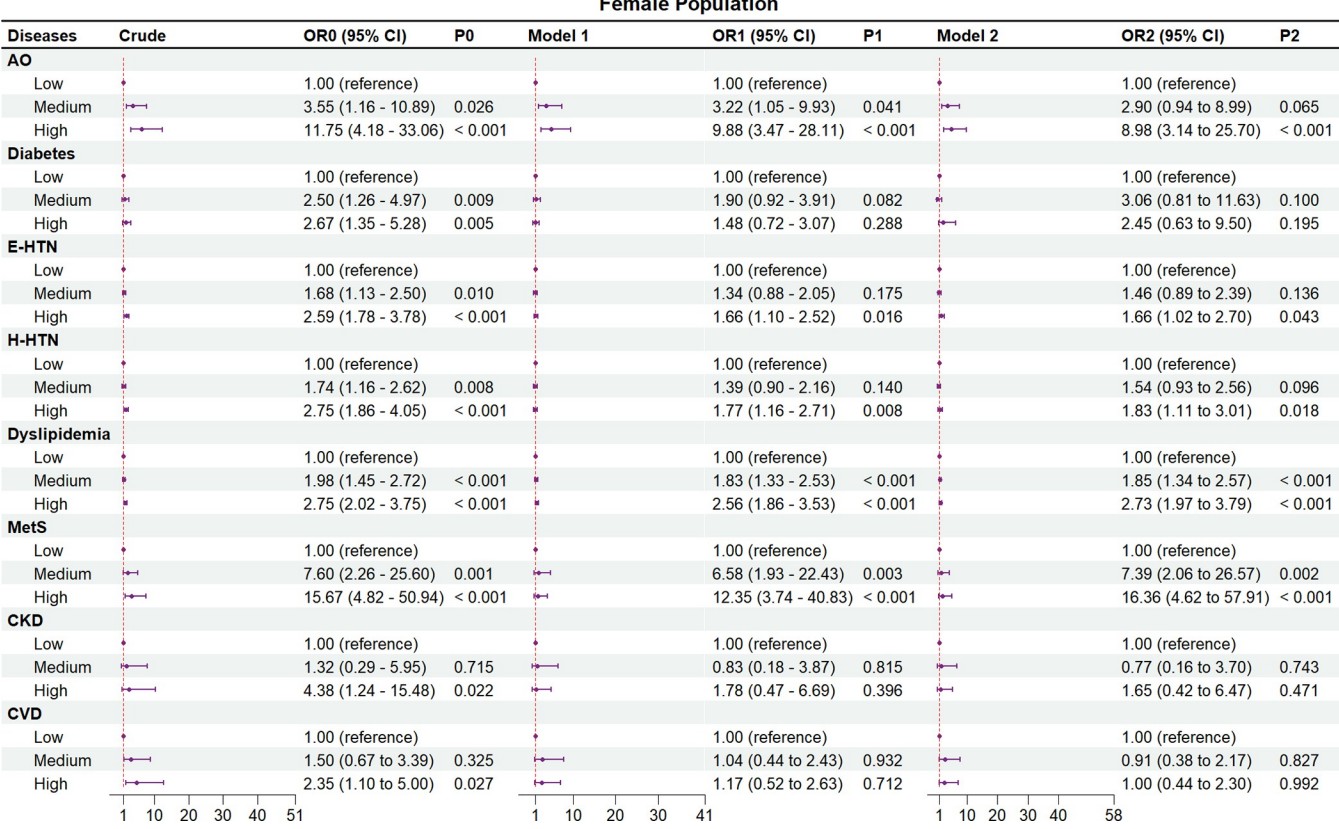

**Female Population**

| Diseases | Crude | OR0 (95% CI) | P0 | Model 1 | OR1 (95% CI) | P1 | Model 2 | OR2 (95% CI) | P2 |
|---|---|---|---|---|---|---|---|---|---|
| **AO** | | | | | | | | | |
| Low | | 1.00 (reference) | | | 1.00 (reference) | | | 1.00 (reference) | |
| Medium | | 3.55 (1.16 - 10.89) | 0.026 | | 3.22 (1.05 - 9.93) | 0.041 | | 2.90 (0.94 - 8.99) | 0.065 |
| High | | 11.75 (4.18 - 33.06) | < 0.001 | | 9.88 (3.47 - 28.11) | < 0.001 | | 8.98 (3.14 - 25.70) | < 0.001 |
| **Diabetes** | | | | | | | | | |
| Low | | 1.00 (reference) | | | 1.00 (reference) | | | 1.00 (reference) | |
| Medium | | 2.50 (1.26 - 4.97) | 0.009 | | 1.90 (0.92 - 3.91) | 0.082 | | 3.06 (0.81 - 11.63) | 0.100 |
| High | | 2.67 (1.35 - 5.28) | 0.005 | | 1.48 (0.72 - 3.07) | 0.288 | | 2.45 (0.63 - 9.50) | 0.195 |
| **E-HTN** | | | | | | | | | |
| Low | | 1.00 (reference) | | | 1.00 (reference) | | | 1.00 (reference) | |
| Medium | | 1.68 (1.13 - 2.50) | 0.010 | | 1.34 (0.88 - 2.05) | 0.175 | | 1.46 (0.89 - 2.39) | 0.136 |
| High | | 2.59 (1.78 - 3.78) | < 0.001 | | 1.66 (1.10 - 2.52) | 0.016 | | 1.66 (1.02 - 2.70) | 0.043 |
| **H-HTN** | | | | | | | | | |
| Low | | 1.00 (reference) | | | 1.00 (reference) | | | 1.00 (reference) | |
| Medium | | 1.74 (1.16 - 2.62) | 0.008 | | 1.39 (0.90 - 2.16) | 0.140 | | 1.54 (0.93 - 2.56) | 0.096 |
| High | | 2.75 (1.86 - 4.05) | < 0.001 | | 1.77 (1.16 - 2.71) | 0.008 | | 1.83 (1.11 - 3.01) | 0.018 |
| **Dyslipidemia** | | | | | | | | | |
| Low | | 1.00 (reference) | | | 1.00 (reference) | | | 1.00 (reference) | |
| Medium | | 1.98 (1.45 - 2.72) | < 0.001 | | 1.83 (1.33 - 2.53) | < 0.001 | | 1.85 (1.34 - 2.57) | < 0.001 |
| High | | 2.75 (2.02 - 3.75) | < 0.001 | | 2.56 (1.86 - 3.53) | < 0.001 | | 2.73 (1.97 - 3.79) | < 0.001 |
| **MetS** | | | | | | | | | |
| Low | | 1.00 (reference) | | | 1.00 (reference) | | | 1.00 (reference) | |
| Medium | | 7.60 (2.26 - 25.60) | 0.001 | | 6.58 (1.93 - 22.43) | 0.003 | | 7.39 (2.06 - 26.57) | 0.002 |
| High | | 15.67 (4.82 - 50.94) | < 0.001 | | 12.35 (3.74 - 40.83) | < 0.001 | | 16.36 (4.62 to 57.91) | < 0.001 |
| **CKD** | | | | | | | | | |
| Low | | 1.00 (reference) | | | 1.00 (reference) | | | 1.00 (reference) | |
| Medium | | 1.32 (0.29 - 5.95) | 0.715 | | 0.83 (0.18 - 3.87) | 0.815 | | 0.77 (0.16 - 3.70) | 0.743 |
| High | | 4.38 (1.24 - 15.48) | 0.022 | | 1.78 (0.47 - 6.69) | 0.396 | | 1.65 (0.42 - 6.47) | 0.471 |
| **CVD** | | | | | | | | | |
| Low | | 1.00 (reference) | | | 1.00 (reference) | | | 1.00 (reference) | |
| Medium | | 1.50 (0.67 to 3.39) | 0.325 | | 1.04 (0.44 to 2.43) | 0.932 | | 0.91 (0.38 to 2.17) | 0.827 |
| High | | 2.35 (1.10 to 5.00) | 0.027 | | 1.17 (0.52 to 2.63) | 0.712 | | 1.00 (0.44 to 2.30) | 0.992 |

**Fig 4. Associations of normal weight obesity with comorbidities and events of cardiovascular disease among female subjects.** AO, abdominal obesity; BF%, body fat percentage; CI, confidence interval; CKD, chronic kidney disease; CVD, cardiovascular disease; E-HTN, essential hypertension; H-HTN, H-type hypertension; MetS, metabolic syndrome; OR, odds ratio; Ref, reference. Crude, without adjustment for other risk factors. Model 1, adjusted for age and gender. Model 2, adjusted for age, gender, educational level, ethnic group, and statuses of marriage, smoking, and drinking. In addition, the histories of using antidiabetic, antihypertensive, and lipid-regulating drugs were respectively adjusted for diabetes, E-HTN and H-HTN, and dyslipidemia, and the history of using more than three drugs was adjusted for metabolic syndrome.

subjects, the number of subjects in the CKD group did not meet the analysis conditions therefore we did not explore the associations between CKD and NWO. The risks of AO, E-HTN, H-HTN, dyslipidemia, and metabolic syndrome were significantly higher in the NWO populations in the above stratifications and subgrouping analyses. Whether in males, females, or total populations, the risk of CVD events was not significantly higher in the NWO tertile than in the low BF% tertile. Interestingly, the risks of diabetes in the total and female NWO populations increased, which was not consistent with the results reported above (Table 3).

### 3.5. Dose-response relationships between BF% and CVD comorbidities and events

As shown in Fig 5, the dose-response relationship was evaluated in the RCS model. The results indicate a significant non-linear dose-response association between BF% and the AO, diabetes, E-HTN and H-HTN, dyslipidemia, metabolic syndrome, CKD, and CVD events after adjusting for all the studied confounding factors. When the ORs of CVD comorbidities and events were 1.0, the BF% value was approximately 28.60%.

**Table 3. Adjusted associations of normal weight obesity (NWO) with comorbidities and events of cardio-cerebrovascular disease (CVD).**

| Diseases | Low BF% tertile | NWO * | | |
|---|---|---|---|---|
| | | Total | Male | Female |
| AO | | | | |
| OR (95% CI) | Ref 1.00 | 5.64 (3.08–10.34) | 8.41 (2.89–24.52) | 4.22 (2.00–8.92) |
| *P* value | | < 0.001 | < 0.001 | < 0.001 |
| Diabetes | | | | |
| OR (95% CI) | Ref 1.00 | 6.95 (1.60–30.30) | 7.33 (0.76–70.61) | 9.15 (1.17–71.79) |
| *P* value | | 0.010 | 0.085 | 0.035 |
| E-HTN | | | | |
| OR (95% CI) | Ref 1.00 | 1.48 (1.00–2.18) | 1.58 (0.94–2.67) | 1.42 (0.79–2.56) |
| *P* value | | 0.046 | 0.084 | 0.244 |
| H-HTN | | | | |
| OR (95% CI) | Ref 1.00 | 1.55 (1.02–2.35) | 1.75 (1.02–3.00) | 1.35 (0.69–2.64) |
| *P* value | | 0.041 | 0.043 | 0.379 |
| Dyslipidemia | | | | |
| OR (95% CI) | Ref 1.00 | 1.82 (1.46–2.28) | 2.63 (1.88–3.68) | 1.41 (1.03–1.91) |
| *P* value | | < 0.001 | < 0.001 | 0.030 |
| MetS | | | | |
| OR (95% CI) | Ref 1.00 | 9.00 (2.67–30.33) | 28.32 (3.50–228.90) | 3.53 (0.74–16.84) |
| *P* value | | < 0.001 | 0.002 | 0.114 |
| CVD | | | | |
| OR (95% CI) | Ref 1.00 | 0.80 (0.28–2.26) | 0.37 (0.05–2.59) | 1.12 (0.30–4.17) |
| *P* value | | 0.668 | 0.319 | 0.866 |

AO, abdominal obesity; BF%, body fat percentage; CI, confidence interval; E-HTN, essential hypertension; H-HTN, H-type hypertension; MetS, metabolic syndrome; OR, odds ratio; Ref, reference.

Adjusted for age, gender, educational level, ethnic group, and status of marriage, smoking, and drinking.

Individuals who knew they had hypertension, diabetes, and dyslipidemia were excluded.

* NWO was defined as the condition in the high tertile of BF% among the normal body mass index range.

## 4. Discussion

With the increased food production and prevailing sedentary lifestyle, the obesity prevalence in China has increased constantly. Based on the criteria for Chinese populations, more than half of adults were either overweight or obese in the most recent national survey [4]. However, the assessment of obesity in most studies was based on BMI, without considering the influence of BF% and fat distribution. In the study, we only included adults to investigate the association between the increase of BF% and the risk of CVD comorbidities and events with normal body weight.

As the World Health Organization or any major scientific society have not established cut-off values for BF% to define NWO [5], the cut-off values of BF% and definition of NWO vary greatly in different studies, which affects the comparability of results between studies. It is necessary to establish standardized cut-off values for defining NWO in the future. A previous study [36] from China reported that the optimal BF% cut-off in Chinese people was 24% for men and 33% for women, but the data were collected in the 2008–2010 years, about 15 years ago. We used an arbitrary cut-off for BF% based on tertiles to define NWO, which could better reflect the situation of the included population. The prevalence of NWO was 16.36%, higher than the previously reported prevalence of 7.39% in another Chinese population [20], indicating an increase in NWO prevalence.

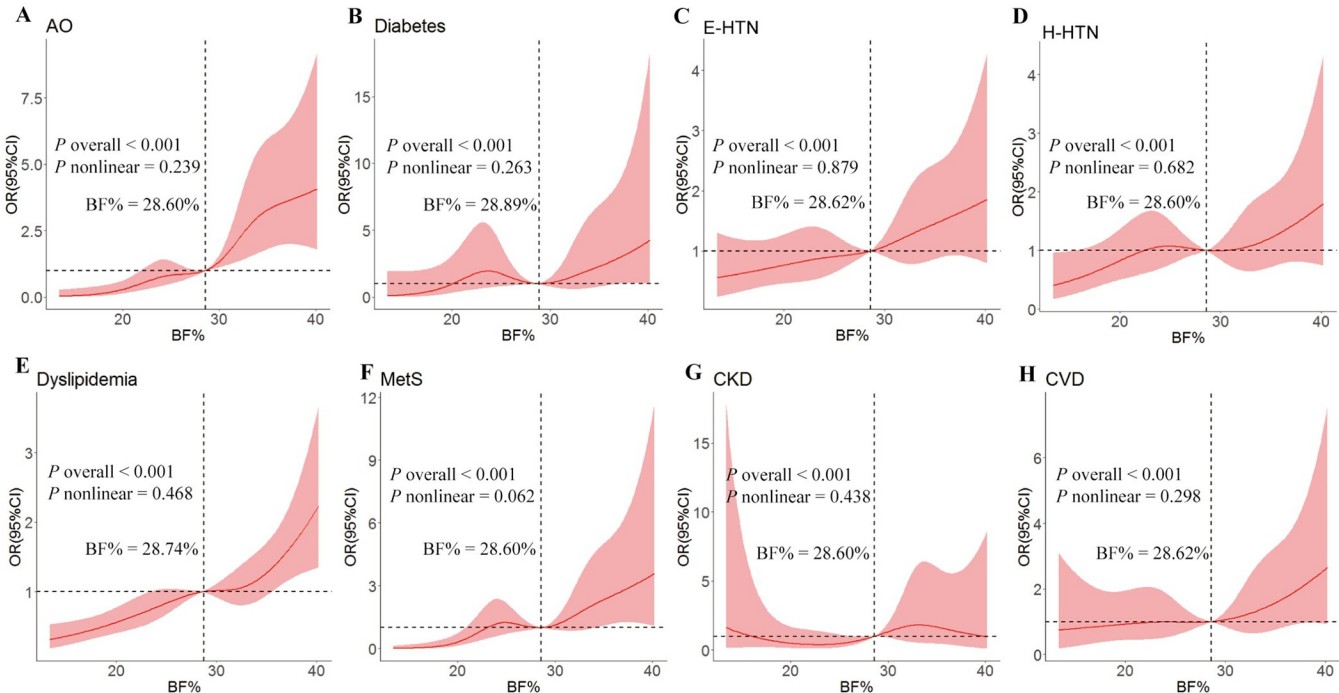

**Fig 5.** The restricted cubic spline for the relationships between BF% and comorbidities (A—G) and events (H) of cardiovascular diseases in the total population. AO, abdominal obesity; BF%, body fat percentage; CKD, chronic kidney disease; CVD, cardiovascular disease; E-HTN, essential hypertension; H-HTN, H-type hypertension; MetS, metabolic syndrome. Three nodes were selected for the model with adjustment for age, gender, educational level, ethnic group, and statuses of marriage, smoking, and drinking. In addition, the histories of antidiabetic, antihypertensive, and lipid-regulating drugs were respectively adjusted for diabetes, E-HTN and H-HTN, and dyslipidemia, and the history of using more than three drugs was adjusted for metabolic syndrome.

The NWO populations in our study had significantly lower HDL-C and higher WC, SBP, DBP, TC, LDL-C, and TG than the non-NWO populations for both genders. It is well known that excessive body fat, regardless of BMI, is a major risk factor for the development of metabolic diseases such as diabetes, hypertension, and dyslipidemia [37]. In our study, we found a significantly higher proportion of AO, diabetes, E-HTN, H-HTN, and metabolic syndrome in the NWO group when compared to the lowest BF% tertile. After adjustment for concomitant variables (Model 2), NWO was only significantly associated with AO, E-HTN, dyslipidemia, and metabolic syndrome among the total populations, which was in fact, similar to the previous findings [13, 20, 25, 36]. In a national study from China [20], NWO had definite correlations with the risks of hypertension and metabolic syndrome, which suggests that cardiometabolic risks increase in NWO populations. Coelho et al. [38] also reported NWO individuals showed higher risks for developing metabolic syndrome, increased WC, and elevated glucose levels than the healthy BF% individuals. Some studies attempt to explain why individuals with NWO have higher cardiovascular risks. On the one hand, NWO subjects have a higher degree of inflammatory and pro-thrombotic biomarkers such as plasma homocysteine, interleukins, C-reactive protein, and tumor necrosis factor-alpha [15, 19, 25, 39], and inflammation could cause damage to the inflamed site, resulting in metabolic dysregulation, homeostatic alteration and even some diseases (anemia, various tissue damages, malnutrition, and autoimmune diseases) [40]. On the other hand, Park et al. [41] conducted a genome-wide association study involving 49,915 subjects to identify the NWO genetic indices. He found *GCKR*, *ABCB11*, *CDKAL1*, *CDKN2B*, *NT5C2*, and *APOC1* gene were associated with metabolically unhealthy phenotypes in individuals with normal weight but not in those with obesity,

showing that certain genetic polymorphisms may also have an impact on the metabolic health in NWO populations [41].

We found that the male NWO populations had higher risks of AO, H-HTN, dyslipidemia, and metabolic syndrome while the female NWO populations only had higher risks of AO and dyslipidemia after adjusting confounding factors. Whether NWO was associated with the risks of AO, hypertension, dyslipidemia, and metabolic syndrome in a gender-specific manner, previous studies have given different views. Kim et al. (2014) [18] found subjects with NWO had higher prevalence rates of hypertension, metabolic syndrome, and dyslipidemia both in men and women compared with those with normal BF%. Moy et al. [1] also reported higher odds for AO, hypertension, and dyslipidemia were observed in NWO women. However, Kim et al. (2013) [17] found that NWO males had higher risks of hypertension and dyslipidemia, while NWO females only had higher dyslipidemia risk. This finding could, in part, be explained by the fact that the female hormone, estrogen can protect against increased body adiposity/obesity through their effects to suppress appetite and increase energy expenditure, suppress metabolic disorders, reduce endothelial dysfunction, and protect against vascular injury [42–45]. Furthermore, differences in adipose tissue distribution, hormone levels, or metabolic pathways may contribute to these disparities. Males tend to accrue more visceral fat, leading to the classic android body shape, which has been highly correlated with increased cardiovascular risk. In contrast, females accrue more fat in the subcutaneous depot prior to menopause, a feature that affords protection from the negative consequences associated with obesity and metabolic syndrome [45]. Visceral fat is a source of proinflammatory cytokines that contribute to insulin resistance [46], while the accumulation of fat in the subcutaneous depot is an independent predictor of lower cardiovascular and diabetes-related mortality [47]. Of course, further studies with larger sample sizes should conducted to examine the gender based on effect and the difference in body fat distribution in the future.

Interestingly, FBG and HbA1c were not associated with BF%. After adjusting for concomitant variables, we found that NWO was not significantly associated with diabetes in either gender or a specific gender, which was inconsistent with previous findings in China [20, 36]. Xu et al. [36] conducted a 9-year longitudinal survey and showed that Chinese people with NWO had an approximately two times greater risk of developing diabetes compared to normal weight non-obese controls. Jia et al. [20] revealed that the risk of diabetes in the NWO males was higher than that in the normal BF% males, while it was not the case in the females. In addition, a recently published meta [2] reported the risk of diabetes with NWO was higher than that in the normal BF% populations, both in genders. This finding could be explained by the effect of age. Studies have widely recognized the positive relationship between diabetes prevalence and age [48, 49]. Since Shenzhen is a young city with a large of young adults, we found a younger average age of the subjects in the current study, which was 38.5 years. Differences in sample size and study design may also explain this phenomenon. To our interest, after excluding individuals knowing they had diabetes, the risks of diabetes both in the total and female NWO population increased, which may be interpreted as that some patients may actively adopted healthy lifestyles for weight loss and lowering body fat before our investigation, leading to negative results.

There were three studies from China [20, 36, 50] that only reported the association between NWO and cardiometabolic risk factors such as hypertension, diabetes, metabolic syndrome, and dyslipidemia. In the meta-analysis, Khonsari et al. [2] also studied to pool the association between NWO and CVD risk factors, but did not investigate the risk of CVD events in NWO populations. In our study, after adjustment for concomitant variables, we found that the cardiometabolic risks were higher in the NWO populations, but the risk of CVD events in the total NWO populations showed no significant difference from normal BF% subjects. Moreover, few

studies exploring the relationship between NWO and CVD events did not find significance either. Whether in males or females, the risk of CVD events in NWO populations was not higher than that in the normal populations. A previous cohort study by Romero-Corral et al. [25] found that the prevalence of metabolic syndrome and its components increased with the BF% content increased in men and women, and NWO in women was independently associated with increased risk for CVD mortality, but the prevalence of CVD disease did not increase significantly with the BF% increasing both in genders. These negative findings for CVD events in NWO populations could be explained by that we conducted a cross-sectional study and could not establish causality [20], as some CVD patients may actively adopt healthy lifestyles and dietary interventions for weight loss and lowering body fat before the investigation and they were assigned to the group with normal BF%. A Korean study [51] also showed no significant association between NWO and pulse wave velocity or coronary stenosis. Another explanation would be that NWO subjects have higher cardiovascular risk and may be more likely to develop future CVD events and these died from CVD were not surveyed. Because studies reporting the association between NWO and CVD events were limited, large-scale, prospective, and long-term cohort studies are urgently needed to explore whether patients with NWO have a higher risk of future CVD events in the future.

Although obesity is a common metabolic disorder, and there have been numerous studies on conditions that can result from obesity, yet the new concept of obesity (NWO) is not well known or studied [2]. More studies are need to be done so that normal weight obesity gets the recognition it deserves. The public also should be educated on the concept of NWO, and must know that individuals with normal BMI but high BF% also have an increased risk of cardiometabolic conditions. In addition, the screening of NWO might be useful for clinicals in order to implement effective strategies to prevent cardiometabolic diseases [52], guiding healthcare practice to more tailored interventions and preventive strategies for NWO-related health risks.

## Strength and limitations

As far as we know, this is the first study in China to explore the relationship between NWO and CVD events, and we used the high tertile of BF% in the normal BMI range to define NWO, which is more valid than using an arbitrary cut-off not previously validated.

However, our results should be cautiously interpreted for several reasons. First, cross-sectional studies cannot accurately assess causality, and some recall bias may exist. Second, BF% was evaluated by bioelectrical impedance in our study instead of dual-energy X-ray absorptiometry (DXA), which is more accurate in estimating the body fat distribution [53–55]. Third, due to limited budget and time, diabetes was determined by self-reporting and FBG measurement at once, and oral glucose tolerance tests were not performed in our study, which may have caused misclassifications of diabetes patients, resulting in no significant results [40, 41]. Fourth, as there were not enough data in our study to estimate homeostasis model assessment-insulin resistance (HOMA-IR), we did not assess insulin resistance as a cardiometabolic risk factor.

## Conclusions

In conclusion, this study found a relatively high prevalence of NWO among the Chinese population, and cardiometabolic risks such as AO, E-HTN, dyslipidemia, and metabolic syndrome significantly increased in the NWO population. However, these NWO individuals always went unnoticed and undiagnosed due to the limitation of BMI measurement on those with high BF%. It is necessary to carry out NWO screening and active intervention to the public, which can have an important contribution to preventing CVD comorbidities and CVD events.

Though a few studies have explored the relationship between NWO and CVD events, large-scale, prospective, and long-term cohort studies and animal experiments are urgently needed to investigate whether body fat affects the risk of CVD events and the involved mechanisms in the future.

## Supporting information

**S1 File.**
(DOCX)

## Author Contributions

**Conceptualization:** Miaomiao Ma, Deliang Lv, Zhiguang Zhao, Ji-Chang Zhou.

**Data curation:** Xiaobing Wu, Wei Xie, Fengzhu Xie, Qinggang Shang, Ziyang Zhang, Zhiguang Zhao.

**Formal analysis:** Miaomiao Ma, Deliang Lv.

**Investigation:** Xiaobing Wu, Wei Xie, Fengzhu Xie, Qinggang Shang, Ziyang Zhang, Zhiguang Zhao.

**Methodology:** Miaomiao Ma, Xiaobing Wu, Yuqing Chen, Shimiao Dai, Wei Xie, Fengzhu Xie, Qinggang Shang, Ziyang Zhang, Zhiguang Zhao, Ji-Chang Zhou.

**Project administration:** Miaomiao Ma, Xiaobing Wu, Yuqing Chen, Shimiao Dai, Wei Xie, Fengzhu Xie, Qinggang Shang, Ziyang Zhang, Zhiguang Zhao, Ji-Chang Zhou.

**Supervision:** Miaomiao Ma.

**Validation:** Miaomiao Ma, Deliang Lv.

**Writing – original draft:** Miaomiao Ma, Xiaobing Wu, Yuqing Chen, Shimiao Dai, Hui Yang.

**Writing – review & editing:** Miaomiao Ma, Deliang Lv, Xiaobing Wu, Yuqing Chen, Shimiao Dai, Yutian Luo, Hui Yang, Wei Xie, Fengzhu Xie, Qinggang Shang, Ziyang Zhang, Zhiguang Zhao, Ji-Chang Zhou.

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
