## [Decision Letter · Decision Letter 0]

23 May 2024

PONE-D-24-05609Association between normal weight obesity and comorbidities and events of cardiovascular diseases among adults in South ChinaPLOS ONE

Dear Dr. Zhou,

Thank you for submitting your manuscript to PLOS ONE. After careful consideration, we feel that it has merit but does not fully meet PLOS ONE’s publication criteria as it currently stands. Therefore, we invite you to submit a revised version of the manuscript that addresses the points raised during the review process.

We look forward to receiving your revised manuscript.

Kind regards,

Tariq Jamal Siddiqi

Academic Editor

PLOS ONE

“This study was funded by the Sanming Project of Medicine in Shenzhen (SZSM202311019), the Shenzhen Medical Key Discipline Construction Fund (No. SZXK065), the Shenzhen Science and Technology Innovation Committee (No. JCYJ20210324121600003), the National Natural Science Foundation of China (81973038), and the Science, Technology and Innovation Commission of Shenzhen Municipality, China (JCYJ20200109142446804).”

“This study was funded by the Sanming Project of Medicine in Shenzhen (SZSM202311019), the Shenzhen Medical Key Discipline Construction Fund (No. SZXK065), the Shenzhen Science and Technology Innovation Committee (No. JCYJ20210324121600003), the National Natural Science Foundation of China (81973038), and the Science, Technology and Innovation Commission of Shenzhen Municipality, China (JCYJ20200109142446804).”

“This study was funded by the Sanming Project of Medicine in Shenzhen (SZSM202311019), the Shenzhen Medical Key Discipline Construction Fund (No. SZXK065), the Shenzhen Science and Technology Innovation Committee (No. JCYJ20210324121600003), the National Natural Science Foundation of China (81973038), and the Science, Technology and Innovation Commission of Shenzhen Municipality, China (JCYJ20200109142446804).”

Reviewers' comments:

Reviewer's Responses to Questions

**Comments to the Author**

1. Is the manuscript technically sound, and do the data support the conclusions?

Reviewer #1: Yes

Reviewer #2: Yes

2. Has the statistical analysis been performed appropriately and rigorously? 

Reviewer #1: Yes

Reviewer #2: Yes

3. Have the authors made all data underlying the findings in their manuscript fully available?

Reviewer #1: Yes

Reviewer #2: Yes

4. Is the manuscript presented in an intelligible fashion and written in standard English?

Reviewer #1: Yes

Reviewer #2: Yes

5. Review Comments to the Author

Reviewer #1: In their study titled “Association between normal weight obesity and comorbidities and events of cardiovascular diseases among adults in South China,” Zhou et al. aimed to investigate the association between normal weight obesity and the risk of cardiovascular diseases. In my opinion, the following suggestions can enhance the quality of the manuscript:

1. The authors may consider replacing all occurrences of MeTs with metabolic syndrome throughout the entire manuscript, as using the acronym may cause confusion for readers.

2. Abstract, methods section: For transparency, it is kindly advised that the authors briefly outline all the statistical tests conducted in this study.

3. Abstract, methods section line 29: The authors may consider replacing 'by' with 'for' in the phrase “adjusted by multiple confounders” for greater clarity and precision.

4. Main manuscript, introduction section, lines 57-58: the authors may consider rephrasing the sentence “patients with high BF% at normal BMI 57 status can also be risk factors for cardiovascular diseases (CVD)” as “patients with a high BF% despite having a BMI can also be at risk for cardiovascular diseases (CVD).” for greater clarity

5. Main manuscript, introduction section, lines 61-62: although the authors have referenced a meta-analysis indicating the correlation between NWO, dyslipidemia and metabolic syndrome, providing additional details such as the lead author's name, the meta-analysis objectives, and the number of studies/patients involved in these outcomes would enhance the manuscript's transparency.

6. Main manuscript, introduction section, lines 71-73: while the authors have stated and also referenced, that some studies found a worse prognosis with regards to CVD in leaner patients, it would add to the quality of the manuscript, if the authors consider briefly mentioning the details of one such study such as, the title (if it is a clinical trial), lead author's name, main outcomes, and the number of participants etc.

7. To enhance clarity for readers, it is suggested that all instances of "PA" be replaced with "physical activity" throughout the manuscript. Avoiding redundant acronyms can contribute to better understanding of the text.

8. Main manuscript, methods section, measurements portion line 104: the full form of BP along with the acronym should be mentioned as this the first instance this acronym has appeared in the manuscript.

9. Main manuscript, methods section, measurements portion line 117: The authors have mistakenly written “FPG” instead of “FBG” and should make the necessary corrections throughout the manuscript.

10. For better clarity and readability the authors should consider replacing all instances of E-H and H-H with E-HTN and H-HTN respectively throughout the manuscript.

11. Main manuscript, methods section, statistical analysis portion lines 190-191: the sentence can be refined for better clarity as follows "Two models were proposed: Model 1 was adjusted for age and gender, while Model 2 was adjusted for potential confounders identified through Spearman correlation analysis."

12. Main manuscript, results section line 212-213: For better readability the authors are kindly advised to rephrase this sentence as follows: "Among the 3,086 subjects with normal BMI, 39.47% were male. The NWO population consisted of individuals from either gender who were included in the high tertile of BF%."

13. Main manuscript, results section line 352: For increased precision, the authors should consider replacing "whether in Model 1 or Model 2" with "either in Model 1 or Model 2."

14. Main manuscript, results section line 352-356: These lines could be refined for better clarity as follows: "Furthermore, both the male (OR1 = 1.17, 95%CI1 = 0.52 - 2.63; OR2 = 1.00, 95%CI2 = 0.44 - 2.30) and female (OR1 = 2.56, 95%CI1 = 0.93 - 7.05; OR2 = 2.53, 95%CI2 = 0.91 - 7.06) subgroups exhibited similar findings to the overall populations, indicating that the risk of CVD events was not significantly higher in the high BF% tertile compared to the lower BF% tertiles."

Reviewer #2: Zhou et al. conducted a cross-sectional study on "Association between normal weight obesity and comorbidities and events of cardiovascular diseases among adults in South China" and concluded that prevalence of NWO among the Chinese population is high. Moreover, they observed significant increases in cardiometabolic risks such as abdominal obesity, essential hypertension, dyslipidemia, and metabolic syndrome in the NWO population. However, addressing the following comments can further improve the manuscript:

1. In reference to lines 46-47 of the introduction, it is important to clarify that the BMI range mentioned (18.5 to 23.9 kg/m2) corresponds to what is considered a normal BMI for Chinese populations. This distinction is crucial in understanding the context of the study's findings.

2. In lines 48-51, the authors discuss the significant health implications of obesity, including its association with various comorbidities and all-cause mortality. However, to enhance the impact of the rationale of the study, it would be beneficial for the authors to incorporate more compelling statistics that illustrate the profound burden of obesity-related health problems and mortality rates. This helps underscore the urgency and importance of addressing this public health issue.

3. The introduction contains some grammatical inconsistencies that can be addressed for clarity and cohesion. For instance, in lines 54 to 57, the sentence could be revised as follows: "A cross-sectional study by Zeng et al. [7] demonstrated that a high body fat percentage (BF%) is a more precise predictor of cardiometabolic risk factors compared to BMI alone. Furthermore, individuals with high BF% despite maintaining a normal BMI may still face an elevated risk of developing cardiovascular diseases (CVDs) [6, 8, 9]." Similarly, lines 61-62 can be improved: "However, a meta-analysis published in 2022 revealed a correlation between NWO and increased odds of dyslipidemia and MetS [2]." These modifications enhance the coherence and readability of the introduction.

4. While the introduction provides a detailed discussion of the concepts of obesity and NWO, the discussion of the obesity paradox could be more focused to avoid potential confusion, as it introduces a different concept that may not be directly relevant to the study objectives.

5. While the introduction mentions the national survey on obesity in China, it could provide more context on the prevalence of NWO specifically in the Chinese population and its implications for public health.

6. The introduction briefly mentions the need to update data on the relationship between BF% and CVD comorbidities and events in the BMI-defined NWO population in China. However, it could provide a more explicit justification for why this research is important and how it will contribute to addressing gaps in knowledge or improving public health outcomes.

7. In the methods section, means for collecting data on anthropometric measurements, blood pressure, and laboratory tests are well-described. However, there's no mention of calibration procedures for ensuring measurement accuracy. Providing information on the reliability and validity of measurement tools used would strengthen the methodological rigor of the study.

8. In line 117 of the methods section, the abbreviation "FPG" for fasting blood glucose should be corrected to "fasting plasma glucose" or abbreviated as "FBG" for consistency and clarity. Ensuring the appropriate use of abbreviations enhances the readability and comprehension of the manuscript.

9. In the statistical analysis section, there's no mention of handling missing data or assessing model assumptions. Providing information on how missing data were handled and assessing the assumptions of statistical models would enhance the robustness of the study findings.

10. The discussion mentions the variability in BF% cut-off values across different studies. However, it could elaborate more on how this variability affects the comparability of results between studies. Providing suggestions or considerations for establishing standardized cut-off values for defining NWO in future research would enhance this part of the discussion.

11. In lines 429-430, the authors mention the presence of specific genetic polymorphisms in NWO individuals, suggesting a potential link to metabolic complications. However, further elaboration on the nature and significance of these genetic variations in contributing to metabolic complications is warranted. This clarification would enhance the understanding of genetic factors contributing to metabolic health and may guide future research and therapeutic interventions aimed at mitigating metabolic risks in this population.

12. In the discussion, the authors acknowledge gender-specific disparities in the relationship between NWO and cardiometabolic risks. While briefly mentioning the protective role of estrogen against endothelial injury, the discussion could benefit from a more thorough exploration of the underlying reasons behind these gender differences. By delving deeper into potential biological or hormonal mechanisms contributing to these disparities, such as differences in adipose tissue distribution, hormone levels, or metabolic pathways, the authors could provide valuable insights into the association between gender and metabolic health.

13. While the discussion effectively summarizes the findings and their underlying factors, adding a paragraph on the clinical implications of these discoveries could enrich it, guiding healthcare practice to more tailored interventions and preventive strategies for NWO-related health risks. Such insights could lead to more targeted screening programs and proactive health management approaches, ultimately contributing to better outcomes and improved public health.

14. Overall, the manuscript contains several grammatical errors, structural inconsistencies, and punctuation errors. Correction of these errors will enhance the clarity and readability of the document, contributing to the overall quality and professionalism of the manuscript.

6. PLOS authors have the option to publish the peer review history of their article (what does this mean?). If published, this will include your full peer review and any attached files.

Reviewer #1: No

Reviewer #2: No

---

## [Author Response · Author response to Decision Letter 0]

26 Jun 2024

Responses are in blue fonts.

Requirements:

When completing the data availability statement of the submission form, you indicated that you will make your data available on acceptance. We strongly recommend all authors decide on a data sharing plan before acceptance, as the process can be lengthy and hold up publication timelines. Please note that, though access restrictions are acceptable now, your entire data will need to be made freely accessible if your manuscript is accepted for publication. This policy applies to all data except where public deposition would breach compliance with the protocol approved by your research ethics board. If you are unable to adhere to our open data policy, please kindly revise your statement to explain your reasoning and we will seek the editor's input on an exemption. Please be assured that, once you have provided your new statement, the assessment of your exemption will not hold up the peer review process.

Answer: 

Thank you for the suggestion. We unable to adhere to your open data policy. When conducting ethical review, we did not inform the survey participants that the data would be made freely accessible, which would violate our ethical review policy. 

Response to Reviewers:

Reviewer #1: In their study titled “Association between normal weight obesity and comorbidities and events of cardiovascular diseases among adults in South China,” Zhou et al. aimed to investigate the association between normal weight obesity and the risk of cardiovascular diseases. In my opinion, the following suggestions can enhance the quality of the manuscript:

1. The authors may consider replacing all occurrences of MeTs with metabolic syndrome throughout the entire manuscript, as using the acronym may cause confusion for readers.

Answer: Thank you for the suggestion. We have replaced the abbreviation with the full spelling accordingly in the main text. The abbreviation is only used and annotated in figures and tables for space limitation.

2. Abstract, methods section: For transparency, it is kindly advised that the authors briefly outline all the statistical tests conducted in this study.

Answer: We have revised the manuscript according to your suggestion.

3. Abstract, methods section line 29: The authors may consider replacing 'by' with 'for' in the phrase “adjusted by multiple confounders” for greater clarity and precision.

Answer: We have revised it according to your suggestion.

4. Main manuscript, introduction section, lines 57-58: the authors may consider rephrasing the sentence “patients with high BF% at normal BMI 57 status can also be risk factors for cardiovascular diseases (CVD)” as “patients with a high BF% despite having a BMI can also be at risk for cardiovascular diseases (CVD).” for greater clarity

Answer: Thank you for pointing out the expression mistake in the sentence. We have revised the sentence considering both reviewers’ suggestions for it.

5. Main manuscript, introduction section, lines 61-62: although the authors have referenced a meta-analysis indicating the correlation between NWO, dyslipidemia and metabolic syndrome, providing additional details such as the lead author's name, the meta-analysis objectives, and the number of studies/patients involved in these outcomes would enhance the manuscript's transparency.

Answer: We have revised the manuscript according to your suggestion.

6. Main manuscript, introduction section, lines 71-73: while the authors have stated and also referenced, that some studies found a worse prognosis with regards to CVD in leaner patients, it would add to the quality of the manuscript, if the authors consider briefly mentioning the details of one such study such as, the title (if it is a clinical trial), lead author's name, main outcomes, and the number of participants etc.

Answer: We have revised the manuscript according to your suggestion.

7. To enhance clarity for readers, it is suggested that all instances of "PA" be replaced with "physical activity" throughout the manuscript. Avoiding redundant acronyms can contribute to better understanding of the text.

Answer: We have revised the manuscript according to your suggestion.

8. Main manuscript, methods section, measurements portion line 104: the full form of BP along with the acronym should be mentioned as this the first instance this acronym has appeared in the manuscript.

Answer: We have revised it according to your suggestion.

9. Main manuscript, methods section, measurements portion line 117: The authors have mistakenly written “FPG” instead of “FBG” and should make the necessary corrections throughout the manuscript.

Answer: Thank you. We have revised the abbreviation throughout the manuscript according to your suggestion.

10. For better clarity and readability the authors should consider replacing all instances of E-H and H-H with E-HTN and H-HTN respectively throughout the manuscript.

Answer: Thank you for the suggestion. We have revised those two abbreviations accordingly.

11. Main manuscript, methods section, statistical analysis portion lines 190-191: the sentence can be refined for better clarity as follows "Two models were proposed: Model 1 was adjusted for age and gender, while Model 2 was adjusted for potential confounders identified through Spearman correlation analysis."

Answer: We have revised the manuscript according to your suggestion.

12. Main manuscript, results section line 212-213: For better readability the authors are kindly advised to rephrase this sentence as follows: "Among the 3,086 subjects with normal BMI, 39.47% were male. The NWO population consisted of individuals from either gender who were included in the high tertile of BF%."

Answer: We have rephrased the sentence as you suggested.

13. Main manuscript, results section line 352: For increased precision, the authors should consider replacing "whether in Model 1 or Model 2" with "either in Model 1 or Model 2."

Answer: Thank you for kindly pointing out the mistake. We have corrected it according to your suggestion.

14. Main manuscript, results section line 352-356: These lines could be refined for better clarity as follows: "Furthermore, both the male (OR1 = 1.17, 95%CI1 = 0.52 - 2.63; OR2 = 1.00, 95%CI2 = 0.44 - 2.30) and female (OR1 = 2.56, 95%CI1 = 0.93 - 7.05; OR2 = 2.53, 95%CI2 = 0.91 - 7.06) subgroups exhibited similar findings to the overall populations, indicating that the risk of CVD events was not significantly higher in the high BF% tertile compared to the lower BF% tertiles."

Answer: Thank you for the good suggestion. We have revised the sentence as you suggested.

Reviewer #2: Zhou et al. conducted a cross-sectional study on "Association between normal weight obesity and comorbidities and events of cardiovascular diseases among adults in South China" and concluded that prevalence of NWO among the Chinese population is high. Moreover, they observed significant increases in cardiometabolic risks such as abdominal obesity, essential hypertension, dyslipidemia, and metabolic syndrome in the NWO population. However, addressing the following comments can further improve the manuscript:

1. In reference to lines 46-47 of the introduction, it is important to clarify that the BMI range mentioned (18.5 to 23.9 kg/m2) corresponds to what is considered a normal BMI for Chinese populations. This distinction is crucial in understanding the context of the study's findings.

Answer: We have made changes in lines 49-50 of the introduction.

2. In lines 48-51, the authors discuss the significant health implications of obesity, including its association with various comorbidities and all-cause mortality. However, to enhance the impact of the rationale of the study, it would be beneficial for the authors to incorporate more compelling statistics that illustrate the profound burden of obesity-related health problems and mortality rates. This helps underscore the urgency and importance of addressing this public health issue.

Answer: Thank you for the critical suggestion. We have presented the statistics in lines 55-63 of the introduction that illustrate the profound burden of obesity-related health problems.

3. The introduction contains some grammatical inconsistencies that can be addressed for clarity and cohesion. For instance, in lines 54 to 57, the sentence could be revised as follows: "A cross-sectional study by Zeng et al. [7] demonstrated that a high body fat percentage (BF%) is a more precise predictor of cardiometabolic risk factors compared to BMI alone. Furthermore, individuals with high BF% despite maintaining a normal BMI may still face an elevated risk of developing cardiovascular diseases (CVDs) [6, 8, 9]." Similarly, lines 61-62 can be improved: "However, a meta-analysis published in 2022 revealed a correlation between NWO and increased odds of dyslipidemia and MetS [2]." These modifications enhance the coherence and readability of the introduction.

Answer: Thank you very much! We have revised those sentences considering both reviewers’ suggestions.

4. While the introduction provides a detailed discussion of the concepts of obesity and NWO, the discussion of the obesity paradox could be more focused to avoid potential confusion, as it introduces a different concept that may not be directly relevant to the study objectives.

Answer: Thank you! Studies have reported an obesity paradox between cardiovascular disease and obesity, but the relationship between NWO and cardiovascular disease remains unclear. And we aimed to explore the latter relationship.

5. While the introduction mentions the national survey on obesity in China, it could provide more context on the prevalence of NWO specifically in the Chinese population and its implications for public health.

Answer: We have supplemented the data on the prevalence of NWO in the Chinese population in lines 92-94 of the introduction.

6. The introduction briefly mentions the need to update data on the relationship between BF% and CVD comorbidities and events in the BMI-defined NWO population in China. However, it could provide a more explicit justification for why this research is important and how it will contribute to addressing gaps in knowledge or improving public health outcomes.

Answer: We have made changes in lines 84-89 and 90-103 of the introduction section.

7. In the methods section, means for collecting data on anthropometric measurements, blood pressure, and laboratory tests are well-described. However, there's no mention of calibration procedures for ensuring measurement accuracy. Providing information on the reliability and validity of measurement tools used would strengthen the methodological rigor of the study.

Answer: We have provided information on the reliability and validity of measurement tools in lines 135-149 of the methods section.

8. In line 117 of the methods section, the abbreviation "FPG" for fasting blood glucose should be corrected to "fasting plasma glucose" or abbreviated as "FBG" for consistency and clarity. Ensuring the appropriate use of abbreviations enhances the readability and comprehension of the manuscript.

Answer: We have revised the abbreviation “FPG” in the whole text.

9. In the statistical analysis section, there's no mention of handling missing data or assessing model assumptions. Providing information on how missing data were handled and assessing the assumptions of statistical models would enhance the robustness of the study findings.

Answer: We mentioned in the results that 63 individuals without BMI and BF% data were excluded from the analysis. A total of 3,086 subjects with BMI and BF% included for analysis, of which 189 patients lacked lipid level data and 7 patients lacked FBG data. When performing Logistic regression, records containing missing values were excluded.

10. The discussion mentions the variability in BF% cut-off values across different studies. However, it could elaborate more on how this variability affects the comparability of results between studies. Providing suggestions or considerations for establishing standardized cut-off values for defining NWO in future research would enhance this part of the discussion.

Answer: We have provided suggestions for establishing standardized cut-off values for defining NWO in future in lines 438-441 of the discussion section.

11. In lines 429-430, the authors mention the presence of specific genetic polymorphisms in NWO individuals, suggesting a potential link to metabolic complications. However, further elaboration on the nature and significance of these genetic variations in contributing to metabolic complications is warranted. This clarification would enhance the understanding of genetic factors contributing to metabolic health and may guide future research and therapeutic interventions aimed at mitigating metabolic risks in this population.

Answer: We have added further elaboration in lines 467-473 of the discussion section.

12. In the discussion, the authors acknowledge gender-specific disparities in the relationship between NWO and cardiometabolic risks. While briefly mentioning the protective role of estrogen against endothelial injury, the discussion could benefit from a more thorough exploration of the underlying reasons behind these gender differences. By delving deeper into potential biological or hormonal mechanisms contributing to these disparities, such as differences in adipose tissue distribution, hormone levels, or metabolic pathways, the authors could provide valuable insights into the association between gender and metabolic health.

Answer: We have added further elaboration in lines 484-499 of the discussion section.

13. While the discussion effectively summarizes the findings and their underlying factors, adding a paragraph on the clinical implications of these discoveries could enrich it, guiding healthcare practice to more tailored interventions and preventive strategies for NWO-related health risks. Such insights could lead to more targeted screening programs and proactive health management approaches, ultimately contributing to better outcomes and improved public health.

Answer: We have added a paragraph on the clinical implications in lines 544-552 of the discussion section.

14. Overall, the manuscript contains several grammatical errors, structural inconsistencies, and punctuation errors. Correction of these errors will enhance the clarity and readability of the document, contributing to the overall quality and professionalism of the manuscript.

Answer: Thank you for your comments. We’ve tried to fix those issues to our best in the revised version of the manuscript.

---

## [Decision Letter · Decision Letter 1]

7 Aug 2024

PONE-D-24-05609R1Association between normal weight obesity and comorbidities and events of cardiovascular diseases among adults in South ChinaPLOS ONE

Dear Dr. Zhou,

Thank you for submitting your manuscript to PLOS ONE. After careful consideration, we feel that it has merit but does not fully meet PLOS ONE’s publication criteria as it currently stands. Therefore, we invite you to submit a revised version of the manuscript that addresses the points raised during the review process.

We look forward to receiving your revised manuscript.

Kind regards,

Tariq Jamal Siddiqi

Academic Editor

PLOS ONE

Journal Requirements:

Reviewers' comments:

Reviewer's Responses to Questions

**Comments to the Author**

1. If the authors have adequately addressed your comments raised in a previous round of review and you feel that this manuscript is now acceptable for publication, you may indicate that here to bypass the “Comments to the Author” section, enter your conflict of interest statement in the “Confidential to Editor” section, and submit your "Accept" recommendation.

Reviewer #1: All comments have been addressed

Reviewer #2: All comments have been addressed

2. Is the manuscript technically sound, and do the data support the conclusions?

Reviewer #1: Yes

Reviewer #2: Yes

3. Has the statistical analysis been performed appropriately and rigorously? 

Reviewer #1: Yes

Reviewer #2: Yes

4. Have the authors made all data underlying the findings in their manuscript fully available?

Reviewer #1: Yes

Reviewer #2: Yes

5. Is the manuscript presented in an intelligible fashion and written in standard English?

Reviewer #1: Yes

Reviewer #2: Yes

6. Review Comments to the Author

**Reviewer #1: **The authors have addressed all of the comments adequately. The manuscript is now ready for publication.

**Reviewer #2:** I appreciate the authors' responsiveness to my comments; the manuscript has improved. However, there are still several grammatical and structural inconsistencies. To enhance clarity and coherence, I recommend using proof-reading tools to identify and rectify these issues, ensuring the manuscript attains a more polished and professional appearance. Here are a few examples that need revision for better clarity and coherence of the text:

1.Lines 84-88 could be rewritten as “He found diastolic blood pressure and odds of hypertension was significantly raised in the NWO individuals compared with normal weight lean. In addition, adults with NWO had elevated blood glucose and increased odds of high blood sugar levels, highlighting the association between NWO and cardiometabolic

derangements.”

2.Lines 88-90 should be revised to “"People with NWO are likely to develop dyslipidemia [5], insulin resistance [16, 17], changes in blood pressure (BP) [16, 18], and a pro-oxidative status [19]."

3.In line 95, remove the word ‘in addition’ and start the paragraph with ‘there were few epidemiological studies…’

4.Lines 494-499 should be rewritten to “Park et al. [41] conducted a genome-wide association study involving 49,915 subjects to identify the NWO genetic indices. He found GCKR, ABCB11, CDKAL1, CDKN2B, NT5C2, and APOC1 gene were associated with metabolically unhealthy phenotypes in individuals with normal weight but not in those with obesity, showing that certain genetic polymorphisms may also have an impact on the metabolic health in NWO populations.”

5.Lines 518-524 can be rephrased to “Males tend to accrue more visceral fat, leading to the classic android body shape, which has been highly correlated with increased cardiovascular risk. In contrast, females accrue more fat in the subcutaneous depot prior to menopause, a feature that affords protection from the negative consequences associated with obesity and metabolic syndrome [45]. Visceral fat is a source of proinflammatory cytokines that contribute to insulin resistance [46], while the accumulation of fat in the subcutaneous depot is an independent predictor of lower cardiovascular and diabetes-related mortality [47].”

6.Lines 527-530 could be revised to “Interestingly, FPG, FBG, and HbA1c were not associated with BF%. After adjusting for concomitant variables, we found that NWO was not significantly associated with diabetes in either gender or a specific gender, which was inconsistent with previous findings in China [20, 36]”.

7. PLOS authors have the option to publish the peer review history of their article (what does this mean?). If published, this will include your full peer review and any attached files.

Reviewer #1: No

Reviewer #2: No

---

## [Author Response · Author response to Decision Letter 1]

1 Sep 2024

Requirements:

Answer: We have reviewed our reference list, and ensured that it was complete and correct.

Response to Reviewers:

Reviewer #2: I appreciate the authors' responsiveness to my comments; the manuscript has improved. However, there are still several grammatical and structural inconsistencies. To enhance clarity and coherence, I recommend using proof-reading tools to identify and rectify these issues, ensuring the manuscript attains a more polished and professional appearance. Here are a few examples that need revision for better clarity and coherence of the text:

1.Lines 84-88 could be rewritten as “He found diastolic blood pressure and odds of hypertension was significantly raised in the NWO individuals compared with normal weight lean. In addition, adults with NWO had elevated blood glucose and increased odds of high blood sugar levels, highlighting the association between NWO and cardiometabolic derangements.”

Answer: We have made changes in lines 84-88 (current lines of 78-83) of the introduction.

2.Lines 88-90 should be revised to “"People with NWO are likely to develop dyslipidemia [5], insulin resistance [16, 17], changes in blood pressure (BP) [16, 18], and a pro-oxidative status [19]."

Answer: We have made changes in lines 88-90 (current lines of 83-85) of the introduction.

3.In line 95, remove the word ‘in addition’ and start the paragraph with ‘there were few epidemiological studies…’

Answer: We have removed the word ‘in addition’ in line 95 (current line of 91) of the introduction.

4.Lines 494-499 should be rewritten to “Park et al. [41] conducted a genome-wide association study involving 49,915 subjects to identify the NWO genetic indices. He found GCKR, ABCB11, CDKAL1, CDKN2B, NT5C2, and APOC1 gene were associated with metabolically unhealthy phenotypes in individuals with normal weight but not in those with obesity, showing that certain genetic polymorphisms may also have an impact on the metabolic health in NWO populations.”

Answer: We have made changes in lines 494-499 (current lines of 472-474) of the discussion.

5.Lines 518-524 can be rephrased to “Males tend to accrue more visceral fat, leading to the classic android body shape, which has been highly correlated with increased cardiovascular risk. In contrast, females accrue more fat in the subcutaneous depot prior to menopause, a feature that affords protection from the negative consequences associated with obesity and metabolic syndrome [45]. Visceral fat is a source of proinflammatory cytokines that contribute to insulin resistance [46], while the accumulation of fat in the subcutaneous depot is an independent predictor of lower cardiovascular and diabetes-related mortality [47].”

Answer: We have made changes in lines 518-524 (current lines of 491-500) of the discussion.

6.Lines 527-530 could be revised to “Interestingly, FPG, FBG, and HbA1c were not associated with BF%. After adjusting for concomitant variables, we found that NWO was not significantly associated with diabetes in either gender or a specific gender, which was inconsistent with previous findings in China [20, 36]”.

Answer: We have made changes in lines 527-530 (current lines of 501-504) of the discussion.

While revising your submission, please upload your figure files to the Preflight Analysis and Conversion Engine (PACE) digital diagnostic tool, https://pacev2.apexcovantage.com/. PACE helps ensure that figures meet PLOS requirements. To use PACE, you must first register as a user. Registration is free. Then, login and navigate to the UPLOAD tab, where you will find detailed instructions on how to use the tool. If you encounter any issues or have any questions when using PACE, please email PLOS at <a href="mailto:figures@plos.org">figures@plos.org. Please note that Supporting Information files do not need this step.

Answer: We have uploaded our figure files to the Preflight Analysis and Conversion Engine (PACE) digital diagnostic tool, and downloaded PACE generated figure files, to ensure our all figures meet PLOS requirements.

---

## [Decision Letter · Decision Letter 2]

4 Nov 2024

PONE-D-24-05609R2Association between normal weight obesity and comorbidities and events of cardiovascular diseases among adults in South ChinaPLOS ONE

Dear Dr. Zhou,

Thank you for submitting your manuscript to PLOS ONE. After careful consideration, we feel that it has merit but does not fully meet PLOS ONE’s publication criteria as it currently stands. Therefore, we invite you to submit a revised version of the manuscript that addresses the points raised during the review process.

**ACADEMIC EDITOR:**

Please insert: 95%CI for results of Table-1, and also insert the figures along your revised manuscript.

We look forward to receiving your revised manuscript.

Kind regards,

Ozra Tabatabaei-Malazy

Academic Editor

PLOS ONE

Journal Requirements:

Reviewers' comments:

Reviewer's Responses to Questions

**Comments to the Author**

1. If the authors have adequately addressed your comments raised in a previous round of review and you feel that this manuscript is now acceptable for publication, you may indicate that here to bypass the “Comments to the Author” section, enter your conflict of interest statement in the “Confidential to Editor” section, and submit your "Accept" recommendation.

Reviewer #2: All comments have been addressed

Reviewer #3: All comments have been addressed

2. Is the manuscript technically sound, and do the data support the conclusions?

Reviewer #2: Yes

Reviewer #3: Yes

3. Has the statistical analysis been performed appropriately and rigorously? 

Reviewer #2: Yes

Reviewer #3: Yes

4. Have the authors made all data underlying the findings in their manuscript fully available?

Reviewer #2: Yes

Reviewer #3: Yes

5. Is the manuscript presented in an intelligible fashion and written in standard English?

Reviewer #2: Yes

Reviewer #3: Yes

6. Review Comments to the Author

Reviewer #2: The authors have thoroughly addressed all comments, resulting in significant improvements to the manuscript. Hence, I would recommend it for acceptance.

Reviewer #3: (No Response)

7. PLOS authors have the option to publish the peer review history of their article (what does this mean?). If published, this will include your full peer review and any attached files.

Reviewer #2: No

Reviewer #3: No

---

## [Author Response · Author response to Decision Letter 2]

5 Dec 2024

Response to Reviewers:

Please insert: 95%CI for results of Table-1, and also insert the figures along your revised manuscript.

Answer: Thank you for making us notice the improper expressions of the data in Table 1. We are sorry for presenting the data in Table 1 confusingly at first glance in the last version. To avoid misunderstanding, we made revisions and described under the table that “Categorical data are expressed as n (%). Continuous data are expressed as mean ± SD for normally distributed data or median [IQR] for non-normally distributed data.” Categorical variables were analyzed by Chi-squared tests, and continuous variables were analyzed by one-way ANOVA. In most studies[1-10], test results for Chi-squared tests or one-way ANOVA were provided only by P values and not have 95%CI.

For the figures, 95%CIs were presented for the corresponding data, and no changes are made this time.

Answer: We have made changes to our financial disclosure and have amended it in our cover letter.

---

## [Editor Report · Decision Letter 3]

10 Dec 2024

Association between normal weight obesity and comorbidities and events of cardiovascular diseases among adults in South China

PONE-D-24-05609R3

Dear Dr. Zhou,

We’re pleased to inform you that your manuscript has been judged scientifically suitable for publication and will be formally accepted for publication once it meets all outstanding technical requirements.

Kind regards,

Ozra Tabatabaei-Malazy

Academic Editor

PLOS ONE
---

## [Editor Report · Acceptance letter]

27 Dec 2024

PONE-D-24-05609R3 

PLOS ONE

Dear Dr. Zhou, 

I'm pleased to inform you that your manuscript has been deemed suitable for publication in PLOS ONE. Congratulations! Your manuscript is now being handed over to our production team.

Kind regards, 

on behalf of

Dr. Ozra Tabatabaei-Malazy 

Academic Editor

PLOS ONE